# AN IMAGE IS WORTH MORE THAN 16×16 PATCHES: EXPLORING TRANSFORMERS ON INDIVIDUAL PIXELS

**Duy-Kien Nguyen**[2]   **Mahmoud Assran**[1]   **Unnat Jain**[1]   **Martin R. Oswald**[2]
**Cees G. M. Snoek**[2]   **Xinlei Chen**[1]

[1]FAIR, Meta AI      [2]University of Amsterdam

## ABSTRACT

This work does not introduce a new method. Instead, we present an interesting finding that questions the necessity of the inductive bias of *locality* in modern computer vision architectures. Concretely, we find that vanilla Transformers can operate by directly treating each individual pixel as a token and achieve highly performant results. This is substantially different from the popular design in Vision Transformer, which maintains the inductive bias from ConvNets towards local neighborhoods (*e.g.* by treating each 16×16 patch as a token). We showcase the effectiveness of pixels-as-tokens across three well-studied computer vision tasks: supervised learning for classification and regression, self-supervised learning via masked autoencoding, and image generation with diffusion models. Although it's computationally less practical to directly operate on individual pixels, we believe the community must be made aware of this surprising piece of knowledge when devising the next generation of neural network architectures for computer vision.

## 1 INTRODUCTION

For computer vision, the deep learning revolution can be characterized as a revolution in *inductive biases*. Learning previously occurred on top of manually crafted features, such as those described in Lowe (2004); Dalal & Triggs (2005), which encoded preconceived notions about useful patterns and structures for specific tasks. In contrast, biases in modern features are no longer predetermined but instead shaped by direct learning from data using predefined model architectures. This paradigm shift's dominance highlights the potential of reducing feature biases to create more versatile and capable systems that excel across a wide range of vision tasks.

Beyond features, model architectures also possess inductive biases. Reducing these biases can potentially facilitate greater unification not only across tasks but also across data modalities. The *Transformer* architecture (Vaswani et al., 2017) serves as a great example. Initially developed to process natural languages, its effectiveness was subsequently demonstrated for images (Dosovitskiy et al., 2021), point clouds (Zhao et al., 2021), codes (Chen et al., 2021a), and many other types of data. Notably, compared to its predecessor in vision – ConvNets (LeCun et al., 1989; He et al., 2016), the Vision Transformer (ViT) (Dosovitskiy et al., 2021) carries much less image-specific inductive bias. Nonetheless, the initial advantage from such biases is quickly offset by more data (and models that have enough capacities to store patterns observed within the data), ultimately becoming *restrictions* that prevent ConvNets from scaling further (Dosovitskiy et al., 2021).

Of course, ViT is not entirely free of inductive bias. It gets rid of the spatial hierarchy in the ConvNet and models multiple scales in a plain architecture. However, for other inductive biases, the removal is merely half-way through: location equivariance still exists in its patch projection layer and all the intermediate blocks;[1] and *locality* – the notion that neighboring pixels are more related than pixels that are farther apart – still exists in its 'patchification' step (that represents an image with 16×16 2D patches) and position embeddings (when they are manually designed). Therefore, a natural question arises: can we *completely eliminate* either or both of these two major inductive biases? In this work, we aim to answer this important question.

Surprisingly, we find locality can indeed be removed. We arrive at this conclusion by directly treating each individual pixel as a token for the Transformer and using position embeddings learned from

---

[1]By 'location equivariance', we refer to the adoption of *weight-sharing* mechanism which ensures that the same weights are applied regardless of spatial locations.

| inductive bias | ConvNet | ViT | our work |
|---|:---:|:---:|:---:|
| spatial hierarchy | ✓ | ✗ | ✗ |
| location equivariance | ✓ | ✓ | ✓ |
| locality | ✓ | ✓ | ✗ |

Table 1: **Major inductive biases for vision.** A ConvNet (LeCun et al., 1989; He et al., 2016) has all three: spatial hierarchy, location equivariance, and *locality*, with neighboring pixels being more related than pixels farther apart. Vision Transformer (ViT) (Dosovitskiy et al., 2021) removes the spatial hierarchy, reduces (but still retains) location equivariance and locality. We investigate the *complete removal* of locality by simply applying Transformers on individual pixels. It works surprisingly well, challenging the mainstream belief that locality is a necessity for vision architectures.

scratch. In this way, we introduce zero priors about the 2D grid structure of images. Interestingly, instead of training divergence or steep performance degeneration, we can obtain *better* results in quality from this setup. The fact that this naïve, locality-free Transformer works so well suggests there are *more* signals Transformers can capture by viewing images as sets of individual pixels, rather than 16×16 patches. This finding challenges the conventional community belief that 'locality is a fundamental inductive bias for vision tasks' (see Tab. 1).

We showcase the effectiveness of Transformer on pixels via three different case studies: (i) supervised learning for object classification, where CIFAR-100 (Krizhevsky, 2009) is used for our main experiments thanks to its default 32×32 input size, but the observation also generalizes well to ImageNet (Deng et al., 2009), fine-grained classification using Oxford-102-Flowers (Nilsback & Zisserman, 2008) and depth estimation via regression using NYU-v2 (Silberman et al., 2012); (ii) self-supervised learning on CIFAR-100 via Masked Autoencoding (MAE) (He et al., 2022) for pre-training, and fine-tuning for classification; and (iii) image generation with diffusion models, where we follow the architecture of the Diffusion Transformer (DiT) (Peebles & Xie, 2023), and study its pixel variant on ImageNet using the latent token space provided by VQGAN (Esser et al., 2021). In all three cases, we find Transformers on pixels exhibit reasonable behaviors, and achieves results better in quality than baselines equipped with the locality inductive bias.

As a related investigation, we also examine the importance of two locality designs (position embedding and patchification) within the standard ViT architecture on ImageNet. For position embedding, we have three options: sin-cos (Vaswani et al., 2017), learned, and none – with sin-cos carrying the locality bias whilst the other two do not. To systematically 'corrupt' the locality bias in patchification, we perform a pixel *permutation* before dividing the input into 256-pixel (akin to a 16×16 patch in ViT) tokens. The permutation is fixed across images, and consists of multiple steps that each swaps a pixel pair within a distance threshold. Our results suggest that patchification imposes a stronger locality prior, and removing both location equivariance and locality is not yet feasible.

Treating each pixel as a token will lead to a sequence length much longer than commonly used for images. This is especially cumbersome as Self-Attention in Transformers demands quadratic computations. Therefore, the contribution of our work is on the finding, *not* on proposing a new method. In practice, patchification is still a simple and effective idea that trades quality for efficiency, and locality is still highly *useful*. Nevertheless, we believe our investigation delivers a clean, compelling message that locality is *not* a necessary inductive bias for vision. We believe this finding will be integrated into the community knowledge when exploring the next generations of vision architectures.

## 2 INDUCTIVE BIAS OF LOCALITY

Here we discuss about the inductive bias of locality in mainstream vision architectures: ConvNets and ViTs. Locality is the notion that neighboring pixels are more related than pixels farther apart.

### 2.1 LOCALITY IN CONVNETS

In a ConvNet, locality is reflected in the *receptive fields* of the features computed in each layer. Intuitively, receptive fields cover all the pixels involved in computing a specific feature, and for ConvNets, the fields are local. Specifically, ConvNets consist of several layers, each having convolutional kernels or pooling operations – both of which are local. The field is progressively expanded as the network stacks more layers, but the window is still locally centered at the location of the pixel.

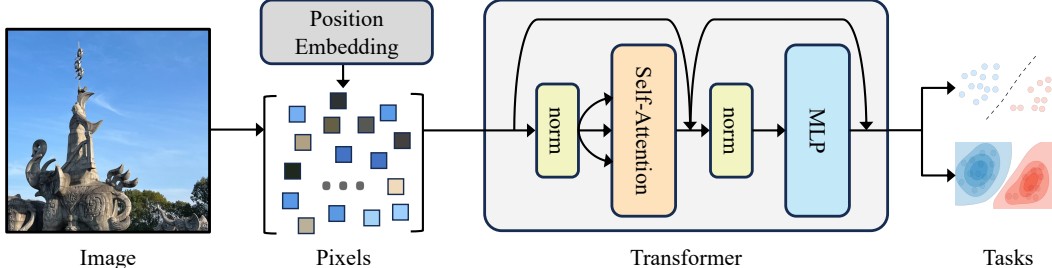

Figure 1: **Transformer on pixels, or 1×1 'patches'**, which we use to investigate the role of locality. Given an image, we simply treat it as a set of pixels. Besides, we also use randomly initialized and learnable position embeddings without prior about the 2D image grid, thus removing the remaining locality bias from previous architectures (*e.g.*, ViT (Dosovitskiy et al., 2014)) that operate on non-degenerated patches. Transformers are employed on the top, with interleaved Self-Attention and MLP blocks (only showing one pair for clarity). We showcase the effectiveness of this *locality-free* architecture through three case studies, spanning both discriminative and generative tasks.

## 2.2 Locality in Vision Transformers

At first glance, Transformers are locality-free. This is because the majority of Transformer operations are either global (*e.g.*, Self-Attention), or purely within each individual token (*e.g.*, MLP). However, a closer look will reveal two designs within ViT (Dosovitskiy et al., 2021) that can still retain the locality inductive bias: patchification and position embedding.

**Locality in patchification.** In ViT, the tokens fed into the Transformer blocks are *patches*, not pixels. Each patch consists of 16×16 pixels, and becomes the basic unit of operation after the first projection layer. This means the amount of computation imposed within the patch is substantially different from the amount across patches: the information outside the 16×16 neighborhood can *only* be propagated with Self-Attention, and the information among the 256 pixels are always processed jointly as token. While the receptive field becomes global after the first Self-Attention block, the bias towards local neighborhoods is already inducted in the patchification step.

**Locality in position embedding.** Position embeddings can be learned (Dosovitskiy et al., 2021), or manually designed and fixed during training. A natural choice for images is to use a 2D sin-cos embedding (Chen et al., 2021b; He et al., 2022), which extends from the original 1D one (Vaswani et al., 2017). As sin-cos functions are smooth, they tend to introduce locality biases that nearby tokens are more similar in the embedding space.[2] Other designed variants are also possible and have been explored (Dosovitskiy et al., 2021), but all of them can carry information about the 2D grid structure of images, unlike learned ones which do not make any assumption about the input.

The locality bias has also been exploited when the position embeddings are *interpolated* (Dehghani et al., 2023; Li et al., 2021). Through bilinear or bicubic interpolation, spatially close embeddings are used to generate a new embedding of the current position, which also leverages locality as a prior.

Compared to ConvNets, ViTs are designed with much less pronounced bias toward locality. This paves the way for our investigation to completely remove this bias, described next.

## 3 Exploring Transformers on Individual Pixels

To study the impact of removing the locality inductive bias, we closely follow the standard Transformer encoder (Vaswani et al., 2017) which processes a sequence of tokens. Particularly, we apply the architecture directly on an unordered set of pixels from an input image with learnable position embeddings. This removes the remaining inductive bias of locality in the ViT (Dosovitskiy et al., 2021) (see Fig. 1). Conceptually, the resulting architecture can be viewed as a simplified version of ViT, with degenerated 1×1 patches instead of non-degenerated ones (*e.g.*, 16×16 or 2×2).

---

[2]While sin-cos functions are also cyclic, it's easy to verify that the majority of their periods are longer than the typical sequence lengths encountered by ViTs.

| model | layers ($N$) | hidden dim ($d$) | MLP dim | heads | param (M) |
|-------|-------|-------|-------|-------|-------|
| T(iny) | 12 | 192 | 768 | 12 | 5.6 |
| S(mall) | 12 | 384 | 1536 | 12 | 21.8 |
| B(ase) | 12 | 768 | 3072 | 12 | 86.0 |
| L(arge) | 24 | 1024 | 4096 | 16 | 303.5 |

Table 2: **Specifications of size variants explored.** We use ViT (Dosovitskiy et al., 2021) or DiT (Peebles & Xie, 2023) with the same size but non-degenerated patches for head-on comparisons.

Formally, we denote the input sequence as $X = (x_1, ..., x_L) \in \mathbb{R}^{L \times d}$, where $L$ is the sequence length and $d$ is the hidden dimension. The Transformer maps the input sequence $X$ to a sequence of representations $Z = (z_1, ..., z_L) \in \mathbb{R}^{L \times d}$. The architecture is a stack of $N$ layers, each of which contains two blocks: multi-headed Self-Attention block and MLP block:

$$\hat{Z}^k = \texttt{SelfAttention}\big(\text{norm}(Z^{k-1})\big) + Z^{k-1},$$
$$Z^k = \texttt{MLP}\big(\text{norm}(\hat{Z}^k)\big) + \hat{Z}^k,$$

where $Z_0$ is the input sequence $X$, $k \in \{1, ..., N\}$ the layer index, and $\text{norm}(\cdot)$ is normalization (*e.g.*, LayerNorm (Ba et al., 2016)). Both blocks use residual connections (He et al., 2016).

**Pixels as tokens.** Given an image $I \in \mathbb{R}^{H \times W \times 3}$ of RGB values, we denote $(H, W)$ as the size of the original input. We follow a simple philosophy and treat $I$ as an unordered set of pixels $(p_l)_{l=1}^{H \cdot W}, p_l \in \mathbb{R}^3$. The first layer simply projects each pixel into a $d$ dimensional vector via linear projection, $f : \mathbb{R}^3 \to \mathbb{R}^d$, resulting in the input set of tokens $X = (f(p_1), ..., f(p_L))$ with $L = H \cdot W$. We learn a content-agnostic position embedding for each position, and optionally append the sequence with a learnable [cls] token (Devlin et al., 2019). The pixel tokens are then fed into the Transformer to produce the set of representations $Z$.

$$X = \big[\texttt{cls}, f(p_1), ..., f(p_L)\big] + \texttt{PE},$$

where $\texttt{PE} \in \mathbb{R}^{L \times d}$ is a set of learnable position embeddings.

Pixel transformer removes the locality inductive bias and is permutation equivariant at the pixel level. By treating individual pixels directly as tokens, we assume no spatial relationship in the architecture and let the model learn structures directly from data. This is in contrast to the design of the convolution kernel in ConvNets or the patch-based tokenization in ViT (Dosovitskiy et al., 2021), which enforces an inductive bias based on the proximity of pixels. In this regard, the resulting model is more versatile – it can naturally model arbitrarily sized images (no need to be divisible by the stride or patch size), or even generalize to irregular regions (Ke et al., 2022).

While our focus is on the study of locality, using each pixel as a separate token has additional benefits. Similar to treating characters as tokens for language, we can greatly reduce the vocabulary size of input tokens to the Transformer. Specifically, given the pixel of RGB channels in the range of $[0, 255]$, the maximum size of vocabulary is $256^3$ (as pixels take integer values); a patch token of size $p \times p$ in ViT, however, can lead to a vocabulary size of up to $256^{3 \cdot p \cdot p}$. If modeled in a non-parametric manner, this will heavily suffer from out-of-vocabulary issues.

Of course, Transformers on pixels are computationally expensive (or even prohibitive). However, given the rapid development of techniques that handle massive sequence lengths for large language models (up to a *million*) (Dao et al., 2022; Liu et al., 2023), it is entirely possible that soon, we can train Transformers on individual pixels directly (*e.g.*, a standard 224×224 crop on ImageNet 'only' contains 50,176 pixels). In this sense, our work is to scientifically verify the effectiveness and potential of locality-free architectures at an affordable scale (which we do next), and leave the engineering effort of scaling for the future.

## 4 EXPERIMENTS

We verify the effectiveness of the locality-free modification with three case studies: supervised learning with ViT (Dosovitskiy et al., 2014), self-supervised learning with MAE (He et al., 2022), and image generation with DiT (Peebles & Xie, 2023). We use four size variants: Tiny (T), Small

| | Acc@1 | Acc@5 |
|---|---|---|
| ViT-T/2 | 83.6 | 94.6 |
| **ViT-T/1** | **85.1** | **96.4** |
| ViT-S/2 | 83.7 | 94.9 |
| **ViT-S/1** | **86.4** | **96.6** |
| ViT-B/2 (Shen et al., 2023) | 72.6 | - |

| | Acc@1 | Acc@5 |
|---|---|---|
| ViT-S/2 | 72.9 | 90.9 |
| **ViT-S/1** | **74.1** | **91.7** |
| ViT-B/2 | 75.7 | 92.3 |
| **ViT-B/1** | **76.1** | **92.6** |
| ViT-L/2 | 75.6 | 92.3 |
| **ViT-L/1** | **76.9** | **93.0** |

(a) **CIFAR-100** classification

(b) **ImageNet** classification

| | Acc@1 | Acc@5 |
|---|---|---|
| ViT-S/2 | 45.8 | 68.3 |
| **ViT-S/1** | **46.3** | **68.9** |

| | RMSE ($\downarrow$) | RAE ($\downarrow$) |
|---|---|---|
| ViT-S/2 | 0.80 | 0.78 |
| **ViT-S/1** | **0.72** | **0.74** |

(c) **Oxford-102-Flower** fine-grained classification

(d) **NYU-v2** depth estimation (regression)

Table 3: **Results for case study #1: supervised learning.** We use ViT (Dosovitskiy et al., 2021), either with non-degenerated patch size ($2\times2$), or with our locality-free modification that treats each pixel as a token (in **bold**). We report results on (a) **CIFAR-100** (Krizhevsky, 2009): the pixel variant *outperforms* ViT with $2\times2$ patches of the same model size. Note that our baselines are already highly-optimized, *e.g.* Shen et al. (2023) reports significantly worse results despite larger model size; the observation generalizes well to the larger (b) **ImageNet** (Deng et al., 2009) dataset, fine-grained classification on (c) **Oxford-102-Flower** (Nilsback & Zisserman, 2008), and depth estimation (regression) on (d) **NYU-v2** (Silberman et al., 2012) which further requires spatial understanding. All these suggest Transformers can not just learn, but learn *well* without any inductive bias of locality.

(S), Base (B) and Large (L) with the specifications shown in Tab. 2. Unless otherwise noted, we use Transformers with the *same* size but *non-degenerated* patches as our baselines. Our *only* goal in this section is to show locality-free architectures can still learn strong vision representations.

## 4.1 CASE STUDY #1: SUPERVISED LEARNING

In this study, we train and evaluate ViTs (Dosovitskiy et al., 2021) with or without locality for supervised learning tasks. Our baselines are ViTs with patch sizes $2\times2$. For more implementation details, please see Appendix A.

**Datasets.** In total we use four datasets, with two for object classification: CIFAR-100 (Krizhevsky, 2009) has 100 classes and 60K images combined, and ImageNet (Deng et al., 2009) has 1K classes and 1.28M images for training, 50K for evaluation. CIFAR-100 is well-suited for exploring the effectiveness of locality-free architectures due to its intrinsic image size of $32 \times 32$. ImageNet has many more images and is more frequently used for modern architecture design. We also validate our finding on: fine-grained classification using the Oxford-102-Flower dataset (Nilsback & Zisserman, 2008) and depth estimation using the NYU-v2 dataset (Silberman et al., 2012).

**Evaluation metrics.** For the classification datasets, we train our models on the `train` split and report the top-1 (Acc@1) and top-5 (Acc@5) accuracy on the `val` split. The root mean squared error (RMSE) and relative absolute error (RAE) are reported on the `val` split for depth estimation.

**Main results.** As shown in Tab. 3, while our baselines for both ViT variants (ViT-T and ViT-S) are well-optimized on CIFAR-100 (*e.g.*, Shen et al. (2023) reports 72.6% Acc@1 when training from scratch with ViT-B, whilst we achieve 80+% with smaller sized models), our locality-free variant, ViT-T/1, improves over ViT-T/2 by 1.5% of Acc@1; and when moving to the bigger model (S), ViT-S/1 leads to an improvement of 1.3% in Acc@1 over ViT-T/1, while ViT/2 starts to saturate. These results suggest compared to the patch-based ViT, ViT with our locality-free design is potentially learning new, data-driven patterns directly from individual pixels.

Our observation also transfers to ImageNet – though with a much lower resolution our results are also lower than the state-of-the-art (Dosovitskiy et al., 2021; Touvron et al., 2021) (80+%), the pixel-based ViT still outperforms patch-based ViT with all model sizes we have experimented, and interestingly synergizes *better* with larger-sized models like ViT-L. To see what the models have learned, we visualize the attention maps, position embeddings in Appendix B.

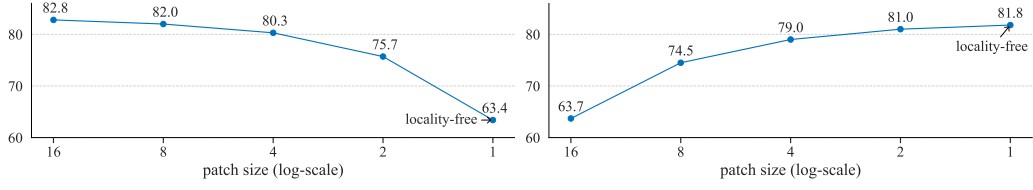

(a) Acc@1 with fixed **sequence length**.       (b) Acc@1 with fixed **input size**.

Figure 2: **Two trends with ViT.** Since our Transformer on pixels can be viewed as ViT with patch size 1×1, the trends w.r.t. patch size is crucial to our finding. In **(a)**, we vary the ViT-B patch size but keep the sequence length fixed (last data point is locality-free) – so the input size is also varied. While Acc@1 remains constant in the beginning, the input size, or the amount of information quickly becomes the dominating factor that deteriorates accuracy. On the other hand, in **(b)** we vary the ViT-S patch size while keeping the input size fixed. The trend is *opposite* – reducing the patch size is always helpful and the last point (*locality-free*) becomes the best. The juxtaposition of these two trends gives a more complete picture of the relationship between input size, patch size and sequence length.

At the expense of longer sequence length, our experiments on fine-grained classification and depth estimation further confirm the effectiveness of the locality-free architecture in boosting accuracies/reducing errors. Notably, fine-grained classification requires nuanced understanding in details; whereas depth estimation demands better capture of spatial relationships. It's impressive that Transformers can learn useful patterns purely from data, *without* assuming inputs are on 2D image grids.

**Two trends with ViT.** With learned position embeddings, our finding is simply based on a ViT variant with 1×1 'patches'. Then why this is not discovered earlier? To us, the major reason is that there are two *different* trends with the three variables in concern: sequence length ($L$), input size ($H{\times}W$) and patch size ($p$). They have a deterministic relationship: $L = H{\times}W/(p^2)$, and can be studied on ImageNet either via fixed sequence length or fixed input size:

- **Fixed *sequence length.*** We show this trend with a fixed $L$ in Fig. 2a. The model size is ViT-B. In this plot, the input size varies (from 224×224 to 14×14) as we vary the patch size (from 16×16 to 1×1). If we follow this trend, then a locality-free architecture (right most) is the *worst*. This means sequence length is not the only deciding factor – input size, or the amount of information fed into the model is arguably a more important factor, especially when it's small. It's only when the size is sufficiently large (*e.g.*, 112×112), the additional benefit of further enlarging the size starts to diminish. This also gives an intuitive explanation that when there is no enough input information, advanced architecture or pre-training (*e.g.*, iGPT (Chen et al., 2020a)) would struggle to recover this hard gap.

- **Fixed *input size.*** Our finding is made when following the other trend – fixing the input size to 64×64 (thus the amount of information), and varying the patch size on ImageNet in Fig. 2b. The model size is ViT-S. Interestingly, we observe an *opposite* trend here: it is always helpful to decrease the patch size (or increase the sequence length), aligned with the prior studies that find sequence length is highly important (Beyer et al., 2022; Hu et al., 2022). Note that the trend holds even when the architecture ultimately becomes locality-free (right-most). With the longest sequence length, it achieves the best accuracy.

Together, the two trends in Fig. 2 *augment* the observations from prior work that mainly focused on sufficiently large input sizes (Beyer et al., 2022; Hu et al., 2022), and give a more complete picture.

## 4.2    CASE STUDY #2: SELF-SUPERVISED LEARNING

Next, we study the impact of removing locality with self-supervised pre-training and then fine-tuning for classification. In particular, we stick to ViT and choose MAE (He et al., 2022) for pre-training, thanks to its efficiency in computation and effectiveness for fine-tuning based evaluation protocols.

**Setup.** We use CIFAR-100 (Krizhevsky, 2009) due to its inherent size of 32×32. This allows us to fully explore the use of pixels as tokens on the original resolution. After pre-training, we test the classification performance on the same set, reporting top-1 (Acc@1) and top-5 (Acc@5) accuracy. For more details, please see Appendix A.

| | pre-train | Acc@1 | Acc@5 |
|---|---|---|---|
| **ViT-T/1** | | 85.1 | 96.4 |
| | ✓ | **86.0** | **97.1** |
| ViT-T/2 | ✓ | 85.7 | 97.0 |

(a) **Tiny**-sized models.

| | pre-train | Acc@1 | Acc@5 |
|---|---|---|---|
| **ViT-S/1** | | 86.4 | 96.6 |
| | ✓ | **87.7** | **97.5** |
| ViT-S/2 | ✓ | 87.4 | 97.3 |

(b) **Small**-sized models.

Table 4: **Results for case study #2: self-supervised learning.** We still use ViT (Dosovitskiy et al., 2021) as the model architecture, but explore MAE (He et al., 2022) as a pre-training technique for enhanced performance. The observation that locality-free variants (in **bold**) work well still holds with pre-training and the two model sizes we experimented.

**Results.** As shown in Tab. 4, the observation that locality-free variants of ViT work well still holds with MAE pre-training and the two model sizes we experimented.

### 4.3 CASE STUDY #3: IMAGE GENERATION

Next, we switch to image generation with a Diffusion Transformer (DiT) (Peebles & Xie, 2023), which has a modulation-based architecture different from the vanilla ViT, and operates on the latent token space from VQGAN (Esser et al., 2021) that shrinks the input size $8\times$. Dataset-wise, we use ImageNet for class-conditional generation, and each image is center-cropped to $256\times256$, resulting in an input feature map size of $32\times32\times4$. More details are found in Appendix A. Our locality-free model, DiT/1, is fed with this feature map, same as its baseline DiT-L/2 with locality.

**Evaluation metrics.** The generation quality is measured by standard metrics: Fréchet Inception Distance (FID) (Heusel et al., 2017) with 50K samples, sFID (Nash et al., 2021), Inception Score (IS) (Salimans et al., 2016), and precision/recall (Kynkäänniemi et al., 2019), using reference batches from the original TensorFlow evaluation suite of Dhariwal & Nichol (2021).

**Qualitative results.** Sampled generations from DiT-L are shown in Fig. 3. The latent diffusion outputs are mapped back to the pixel space using the VQGAN decoder. A classifier-free guidance (Ho & Salimans, 2022) scale of 4.0 is used. For DiT/1, all generations are detailed and reasonable compared to the DiT/2 models with locality.

**Quantitative comparisons.** Tab. 5 summarizes our observations. First, our baseline is strong: compared to the reference 10.67 FID (Peebles & Xie, 2023) with a larger model (DiT-XL/2) and longer training ($\sim$470 epochs), our DiT-L/2 achieves *8.90* without classifier-free guidance. Our main comparison (first two rows) uses a classifier-free guidance of 1.5. DiT-L/1, the locality-free variant, outperforms the baseline on all the 3 main metrics (FID, sFID and IS), and is on-par on precision/recall. With extended training of 1400 epochs, the gap is bigger: 2.68 FID with DiT-L/1 *vs.* 2.89 from DiT-L/2 (see Appendix C).

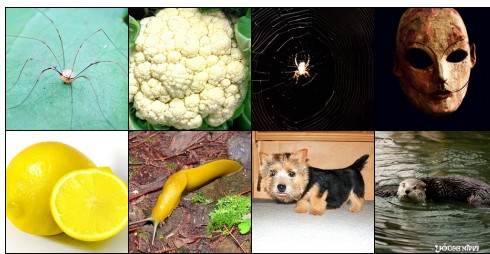
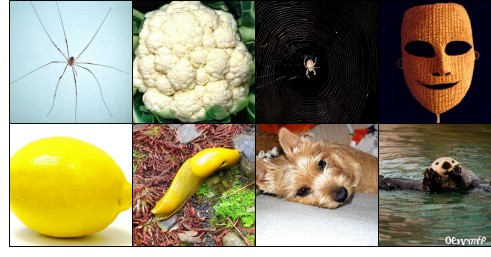

(a) DiT-L/2 generations.        (b) DiT-L/1 generations.

Figure 3: **Qualitative results for case study #3: image generation.** These $256\times256$ samples are generated from ImageNet-trained DiTs (Peebles & Xie, 2023). For direct comparisons, we *fix* random seeds and categories to prompt the model (none of the people classes from ImageNet are used), with the *only* difference that (a) uses locality-biased DiT-L/2, and (b) uses the locality-free variant (**DiT-L/1**). Overall, generations from locality-free models have fine features and detailed and reasonable, similar to locality-based models.

| model | FID (↓) | sFID (↓) | IS (↑) | precision (↑) | recall (↑) |
|---|---|---|---|---|---|
| DiT-L/2 | 4.16 | 4.97 | 210.18 | **0.88** | **0.49** |
| **DiT-L/1** | **4.05** | **4.66** | **232.95** | **0.88** | **0.49** |
| DiT-L/2, no guidance | 8.90 | 4.63 | 104.43 | 0.75 | 0.61 |
| DiT-XL/2 (Peebles & Xie, 2023), no guidance | 10.67 | - | - | - | - |

Table 5: **Results for case study #3: image generation.** We use references from the evaluation suite of Dhariwal & Nichol (2021), and report 5 metrics comparing locality-biased DiT-L/2 and locality-free DiT-L/1. Last row is from Peebles & Xie (2023), compared to which our baseline is significantly stronger (8.90 FID with DiT-L, *vs.* 10.67 with DiT-XL and longer training). Overall, our finding generalizes well to this new task with a different architecture and different input representations.

Our demonstration on image generation is an important extension of our finding. Compared to the case studies on discriminative benchmarks in Sec. 4.1 and Sec. 4.2, the task is more challenging; the model architecture is changed to be conditional; the input space is also changed from raw pixels to latent tokens. The fact that locality-free modification works out-of-the-box suggests the observation can be generalized across different tasks, architectural variants, and operating representations.

## 5 LOCALITY DESIGNS IN VIT

Finally, we complete the loop of our investigation by revisiting the ViT architecture, and examining the importance of its two locality-related designs: (i) position embedding and (ii) patchification.

**Setup.** We use ViT-B for ImageNet supervised classification, adopting the exact same hyper-parameters, augmentations, and other training details from the scratch training recipe of He et al. (2022). Notably, images are crop-and-resized to 224×224 and divided into 16×16 patches.

**Position embedding.** Similar to the investigation in Chen et al. (2021b), we choose from three candidates: sin-cos (Vaswani et al., 2017), learned, and none. The first option introduces locality into the model, while the other two do not. The results are summarized below:

| PE | sin-cos | learned | none |
|---|---|---|---|
| Acc@1 | 82.7 | 82.8 | 81.2 |

Our conclusion is similar to the one drawn by Chen et al. (2021b) for self-supervised representation evaluation: learnable position embeddings are on-par with fixed sin-cos ones. Interestingly, we observe only a minor drop in performance even if there is no position embedding at all – 'none' is only worse by 1.5% compared to sin-cos. Note that without position embedding, the classification model is fully *permutation invariant* w.r.t. patches, though not w.r.t. pixels.

**Patchification.** Next, we use learnable position embeddings and study patchification. To systematically reduce locality from patchification, our key insight is that neighboring pixels should no longer be tied in the same patch. To this end, we perform a pixel-wise permutation before diving the resulting sequence into separate tokens. Each token contains 256 pixels, same in number to pixels in a 16×16 patch. The permutation is shared, staying the same for all the images including testing ones.

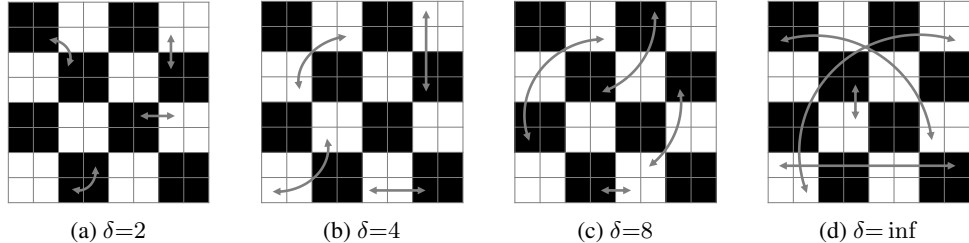

(a) δ=2      (b) δ=4      (c) δ=8      (d) δ= inf

Figure 4: **Pixel permutation for ViT**. We swap pixels within a Hamming distance of $\delta$ and do this $T$ times (no distance constraint if $\delta = \text{inf}$). Illustrated is an 8×8 image divided into 2×2 patches. Here we show permutation with $T = 4$ pixel swaps (denoted by double-headed arrows).

| $T$, $\delta$= inf | Acc@1 | ΔAcc |
|---|---|---|
| 0 | 82.8 | - |
| 100 (0.4%) | 82.1 | -0.7 |
| 1K (4.0%) | 81.9 | -0.9 |
| 10K (39.9%) | 80.5 | -2.3 |
| 20K (79.7%) | 77.5 | -5.3 |
| 25K (99.6%) | 57.6 | -25.2 |

% is the percentage among all pixel pairs

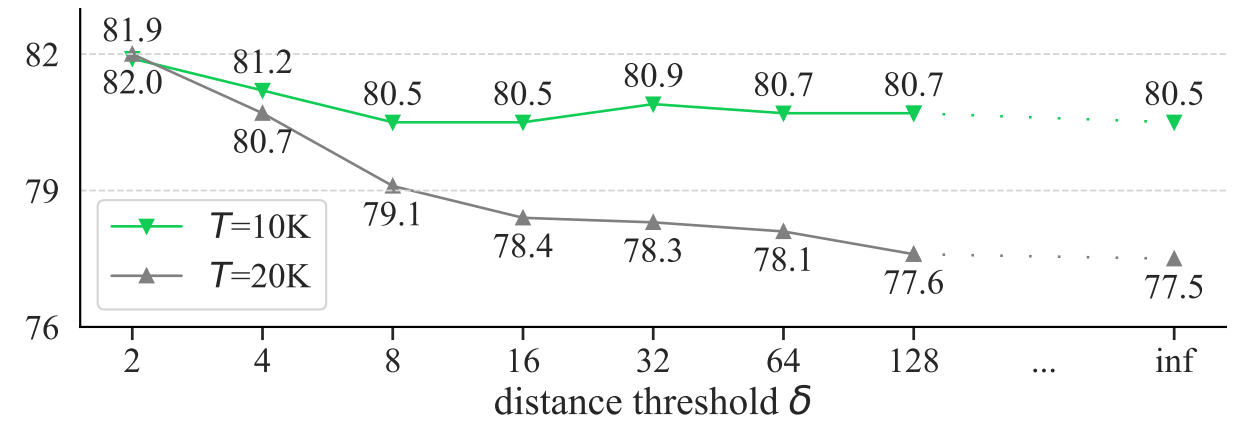

Figure 5: **Results of pixel permutation for ViT-B on ImageNet**. We vary the number of pixel swaps $T$ (*left*) and additionally varying the maximum distance of pixel swaps $\delta$ (*right*). Pixel permutations can drop accuracy by 25.2% (when 25K pairs are swapped), compared to the relatively minor drop (1.6%) when the position embedding is completely removed. And when farther-away pixels are allowed for swapping, more damage is caused. These results suggest pixel permutation imposes a much more significant impact on performance, compared to swapping position embeddings.

The permutation is performed in $T$ steps, each step will swap a pixel pair within a distance threshold $\delta \in [2, \text{inf}]$ (2 means within the 2×2 neighborhood, inf means any pixel pair can be swapped). We use hamming distance on the 2D image grid. $T$ and $\delta$ control how 'corrputed' an image is – larger $T$ or $\delta$ indicates more damage to the local neighborhood and thus more locality bias is taken away. Fig. 4 illustrates four such permutations.

Fig. 5 illustrates the results we have obtained. In the table (left), we vary $T$ with no distance constraint (i.e., $\delta = \text{inf}$). As we increase the number of shuffled pixel pairs, the performance degenerates slowly in the beginning (up to 10K). Then it quickly deteriorates as we further increase $T$. And at $T = 25K$, Acc@1 drops to 57.2%, a 25.2% decrease from the intact image. Note that in total there are $224 \times 224/2 = 25,088$ pixel pairs, so $T = 25K$ means almost all the pixels have moved away from their original location. Fig. 5 (right) shows the influence of $\delta$ given a fixed $T$ (10K or 20K). We can see when farther-away pixels are allowed for swapping (with greater $\delta$), performance gets hurt more. The trend is more salient when more pixel pairs are swapped ($T = 20K$).

Overall, pixel permutation imposes a much more significant impact on Acc@1, compared to changing position embeddings, suggesting that *patchification is much more crucial for the overall design of ViTs*, and underscores the value of our work that removes the patchification altogether.

**Discussion.** As another way to remove locality, pixel permutation is highly destructive. On the other hand, we show successful elimination of locality is possible by treating individual pixels as tokens. We hypothesize this is because permuting pixels not only damages the locality bias, but also hurts the other inductive bias – *location equivariance*. Although locality is removed, the Transformer weights are still shared across pixels to preserve location equivariance; but with shuffling, this inductive bias is also largely removed. The difference suggests that proper weight-sharing remains important and should not be disregarded, especially after locality is already compromised.

## 6 RELATED WORK

**Locality for images.** To of our knowledge, most modern vision architectures (He et al., 2016; Parmar et al., 2018), including those aimed at simplifications of inductive biases (Dosovitskiy et al., 2021; Tolstikhin et al., 2021), still maintain locality in the design. Manual features before deep learning are also locally biased. For example, SIFT (Lowe, 2004) uses a local descriptor to represent each point of interest; HOG (Dalal & Triggs, 2005) locally normalizes the gradient strengths to account for changes in illumination and contrast. Interestingly, with these features, *bag-of-words* models (Csurka et al., 2004; Lazebnik et al., 2006) were popular – analogous to the *set-of-pixels* explored in our work.

**Locality beyond images.** The inductive bias of locality is widely accepted beyond modeling images. For text, a natural language sequence is often pre-processed with 'tokenizers' (Sennrich et al., 2015; Kudo & Richardson, 2018), which aggregate the dataset statistics for grouping frequently-occurring adjacent characters into sub-words. Before Transformers, recurrent neural networks (Mikolov et al., 2010; Hochreiter & Schmidhuber, 1997) were the default architecture for such data, which exploit temporal connectivity to process sequences step-by-step. For even less structured data (*e.g.* point clouds (Chang et al., 2015; Dai et al., 2017)), modern networks (Qi et al., 2017; Zhao et al., 2021) will

resort to various sampling and pooling strategies to increase their sensitivity to the local geometric layout. In graph neural networks (Scarselli et al., 2009), nodes with edges are often viewed as locally connected, and information is propagated through these connections to farther-away nodes. Such designs make them particularly useful for analyzing social networks, molecular structures, *etc*. Despite its higher computational cost, Transformers with global Self-Attention is increasingly favored for real-world problems due to its powerful pattern-learning capabilities.

**Other notable efforts.** We list four efforts in a rough chronological order, and hope it can provide historical context from multiple perspectives for our work:

- To remove locality for ConvNets, Brendel & Bethge (2019) replaced all spatial convolutional filters in a ResNet (He et al., 2016) with 1×1 filters. It has values in interpretability, but without inter-pixel communications the resulting network is substantially worse in performance. Our work instead uses Transformers, which are inherently built on *set* operations, with Self-Attention handling all-to-all communications; understandably, we obtain better results.

- Before ViT gained popularity, iGPT (Chen et al., 2020a) was proposed to directly pre-train Transformers on pixels following their success on text (Radford et al., 2018; Devlin et al., 2019). In retrospect, iGPT is a locality-free model for self-supervised next- (or masked-) pixel prediction. But despite extensive scaling efforts, its performance still falls short compared to simple contrastive pre-training (Chen et al., 2020b) on ImageNet. Later, ViT (Dosovitskiy et al., 2014) *re-introduced* locality (*e.g.*, via patchification) into the architecture, achieving impressive results on many benchmarks including ImageNet. This shift cemented 16×16 patches as the standard for vision tasks. However, it remained unclear whether ViT's performance gains were primarily due to higher resolution or patch-based locality. Through systematic analyses, our work puts a conclusive remark, pointing to resolution as the enabler for ViT, *not* locality.

- Perceiver (Jaegle et al., 2021b;a) is another series of architectures that operate directly on pixels for images. Aimed at being modality-agnostic, Perceiver designs latent Transformers with cross-attention modules to tackle the *efficiency* issue when the input is high-dimensional. However, this design is not as widely adopted as plain Transformers, which have consistently demonstrated scalability across multiple domains (Brown et al., 2020; Dosovitskiy et al., 2021). We show Transformers can indeed work directly with pixels, and given the rapid development of Self-Attention implementations to handle massive sequence length (up to a *million*) (Dao et al., 2022; Liu et al., 2023), efficiency may not be a critical bottleneck even counting all the pixels.

- Transformer has also been shown to work well for distribution modeling of generalized signals, including pixels in Tulsiani & Gupta (2021).

- Our work can also be viewed as extending sequence length scaling to the *extreme* (individual pixels for images). Longer sequence (or higher resolution) is generally beneficial, as evidenced in Chen et al. (2021b); Hu et al. (2022); Peebles & Xie (2023). However, all of them stopped short of reaching the extreme case that completely gets rid of locality.

- Besides individual pixels, there has been sustained interest in learning flexible and data-driven patches for vision models via *grouping* (Zhang & Maire, 2020; Ke et al., 2022; Ma et al., 2023; Deng et al., 2024). While conceptually appealing – *e.g.* can reconcile the long-sequence issue faced with pixels, they are yet to show salient practical promise in this context. Nonetheless, they are related work and we point to this direction for interested readers.

## 7 CONCLUSION AND LIMITATIONS

Through multiple case studies, we have demonstrated that Transformers can directly work with individual pixels. This is surprising, as it allows for a clean, potentially scalable architecture without *locality* – an inductive bias presumably fundamental for vision models. Given the spirit of deep learning that aims to replace manual priors with data-driven, learnable alternatives, we believe our finding is along the right direction and valuable to the community.

However, the practicality and coverage of our current demonstrations remain limited. Given the quadratic computation complexity, treating pixel-based Transformer is more of an approach for investigation, and less for applications. And even with several tasks, the study is still not comprehensive. Nonetheless, we believe this work has sent out a clear, unfiltered message that locality is *not fundamental*, and patchification is simply a *useful* heuristic that trades-off efficiency *vs*. accuracy.

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

## A    IMPLEMENTATION DETAILS

Below, we list the implementation details for the three case studies conducted in Sec. 4.

**Case study #1.** For CIFAR-100, due to the lack of optimal settings even for ViT, we search for the recipe and report results using model sizes T(iny) and S(mall). We use the augmentations from the released demo of He et al. (2020), as we found more advanced augmentations (*e.g.*, AutoAug (Cubuk et al., 2018)) not helpful in this case. All models are trained using AdamW (Loshchilov & Hutter, 2019) with $\beta_1=0.9$, $\beta_2=0.95$. We use a batch size of 1024, weight decay of 0.3, drop path (Huang et al., 2016) of 0.1, and initial learning rate of 0.004 which we found to be the best for all the models. We use a linear learning rate warm-up of 20 epochs and cosine learning rate decay to a minimum of 1e-6. Our training lasts for 2400 epochs, compensating for the small size of the dataset.

On ImageNet, we closely follow the scratch training recipe from Touvron et al. (2021); He et al. (2022) for ViT. Due to our computation budget, images are crop-and-resized to $28\times28$ as the low-resolution inputs by default. The training batch size is 4096, initial learning rate is $1.6\times10^{-3}$, weight decay is 0.3, drop path is 0.1, and training length is 300 epochs. MixUp (Zhang et al., 2018) (0.8), CutMix (Yun et al., 2019) (1.0), RandAug (Cubuk et al., 2020) (9, 0.5), and exponential moving average (0.9999) are used.

For fine-grained classification, we use the same training recipe as in CIFAR-100 with $32\times32$ inputs. For depth estimation, we resize images to the size of $48\times64$ (respecting the original image aspect-ratios) and follow the standard training procedure from Oquab et al. (2023).

**Case study #2.** We follow standard MAE and use a mask ratio of 75% and select tokens randomly. Given the remaining 25% of visible tokens, the model needs to reconstruct masked regions using pixel regression. Since there is no known default setting for MAE on CIFAR-100 (even for ViT), we search for recipes and report results using models of Tiny and Small sizes. The same augmentations as in He et al. (2022) are applied to the images during the pre-training for simplicity. All models are pre-trained using AdamW with $\beta_1=0.9$, $\beta_2=0.95$. We follow all of the hyper-parameters in He et al. (2022) for the pre-training of 1600 epochs except for the initial learning rate of 0.004 and a learning rate decay of 0.85 (Clark et al., 2020). Thanks to MAE pre-training, we can fine-tune our model with a higher learning rate of 0.024. We also set weight decay to 0.02, layer-wise rate decay to 0.65, and drop path to 0.3, $\beta_2$ to 0.999, and fine-tune for 800 epochs. Other hyper-parameters closely follow the scratch training recipe for supervised learning in case study #1.

**Case study #3.** We followed the settings for DiT training, with a larger batch size (2048) to make the training faster (the original recipe uses a batch size of 256). To make the training stable, we perform linear learning rate warm up (Goyal et al., 2017) for 100 epochs and then keep it constant for a total of 400 epochs. We use a maximum learning rate of 8e-4, with no weight decay applied. For generation, a sampling step of 250 is used.

## B    VISUALIZATIONS OF TRANSFORMER ON PIXELS

To check what pixel-based Transformer has learned, we experimented different ways for visualizations. Unless otherwise specified, we use ViT-B and the locality-free variant (*i.e.*, ViT-B/1) trained with supervised learning on ImageNet classification, and compare them side by side.

**Mean attention distances.** In Fig. 6, we present the mean attention distances for ViT/1 and ViT across three categories: late layers (last 4), middle layers (middle 4), and early layers (first 4). Following Dosovitskiy et al. (2021), this metric is computed by aggregating the distances between a query token and all the key tokens in the image space, weighted by their corresponding attention weights. It can be interpreted as *the size of the 'receptive field'* for Transformers. The distance is normalized by the image size, and sorted based on the distance value for different attention heads from left to right.

As shown in Fig. 6a and Fig. 6b, both models exhibit similar patterns in the late layers, with the metric increasing from the 8th to the 11th layer. In the middle layers, while ViT displays a mixed trend among layers (see Fig. 6d), ViT/1 clearly extract patterns from larger areas in the relatively later

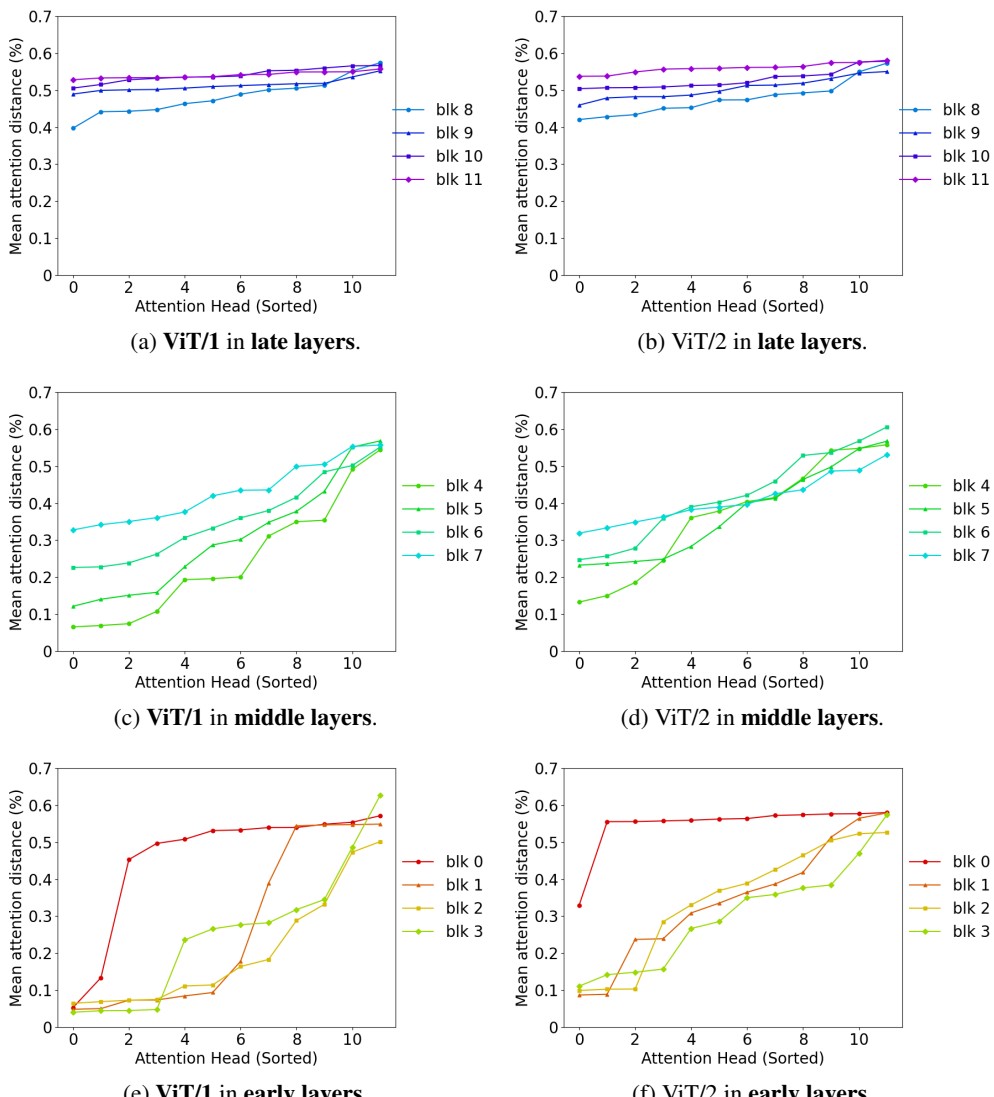

Figure 6: **Mean attention distances** in late, middle, and early layers between ViT/1 and ViT/2. This metric can be interpreted as the receptive field size for Transformers. The distance is normalized by the image size, and sorted based on the distance value for different attention heads from left to right.

layers (see Fig. 6c). Most notably, ViT/1 focuses more on local patterns by paying more attention to small groups of pixels in the early layers, as illustrated in Fig. 6e and Fig. 6f.

**Mean attention offsets.** Fig. 7 shows the mean attention offsets between ViT/1 and ViT as introduced in Walmer et al. (2023). This metric is calculated by determining the center of the attention map generated by a query and measuring the spatial distance (or offset) from the query's location to this center. Thus, the attention offset refers to the degree of spatial deviation of the 'receptive field' – the area of the input that the model focuses on – from the query's original position. Note that different from ConvNets, Self-Attention is a *global* operation, not a local operation that is always centered on the current pixel (offset always being zero).

Interestingly, Fig. 7e suggests that ViT/1 captures long-range relationships in the first layer. Specifically, the attention maps generated by ViT/1 focus on regions far away from the query token – although according to the previous metric (mean attention distance), the overall '*size*' of the attention can be small and focused in this layer.

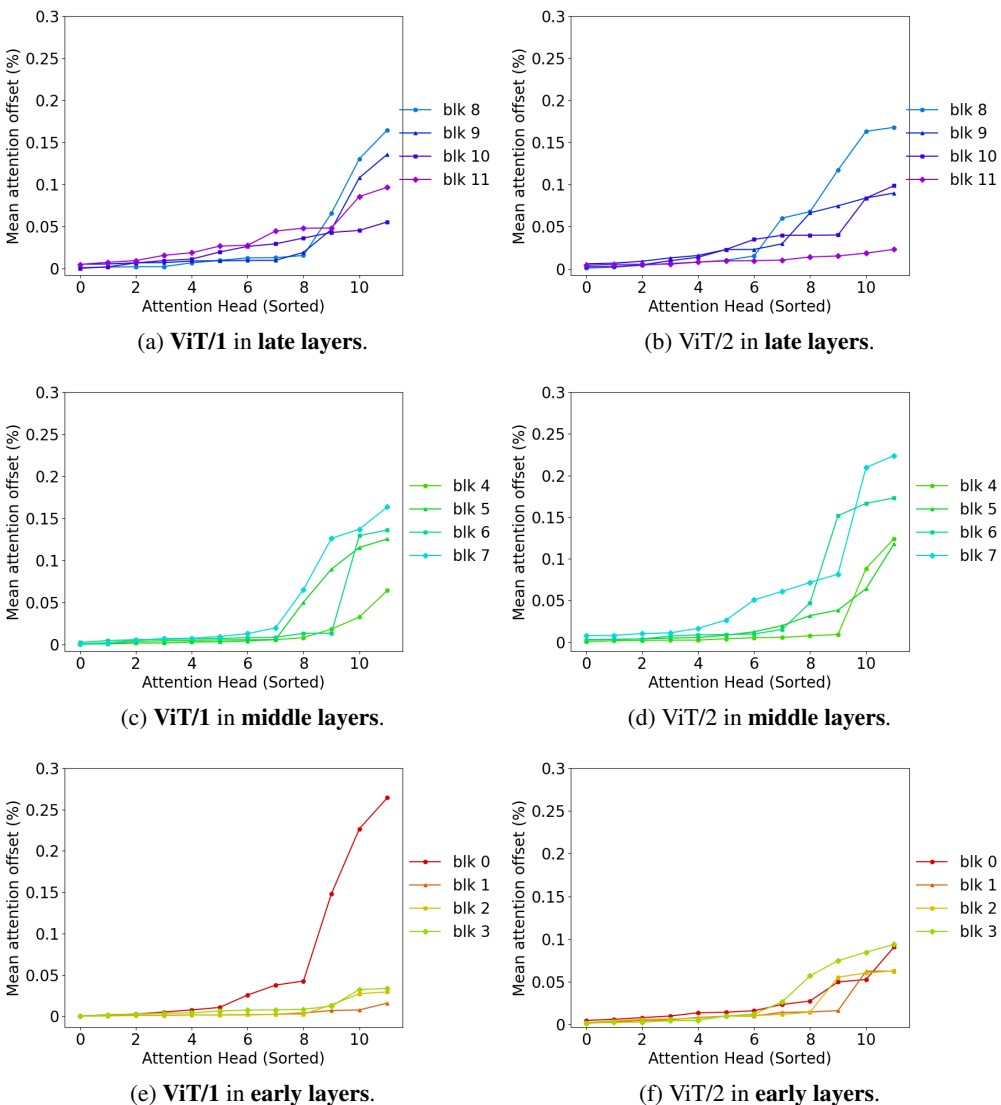

Figure 7: **Mean attention offsets** in late, middle, and early layers between ViT/1 and ViT/2. This metric measures the *deviation* of the attention map from the current token location. The offset is normalized with the image size, and sorted based on the distance value for different attention heads from left to right.

**Figure-ground segmentation in early layers.** In Fig. 8, we observe another interesting behavior of ViT/1. Here, we use the central pixel in the image space as the query and visualize its attention maps in the early layers. We find that the attention maps in the early layers can already capture the *foreground* of objects. Figure-ground segmentation (Boykov et al., 2001) can be effectively performed with low-level signals (*e.g.*, RGB values) and therefore approaches with a few layers. And this separation prepares the model to potentially capture higher-order relationships in later layers.

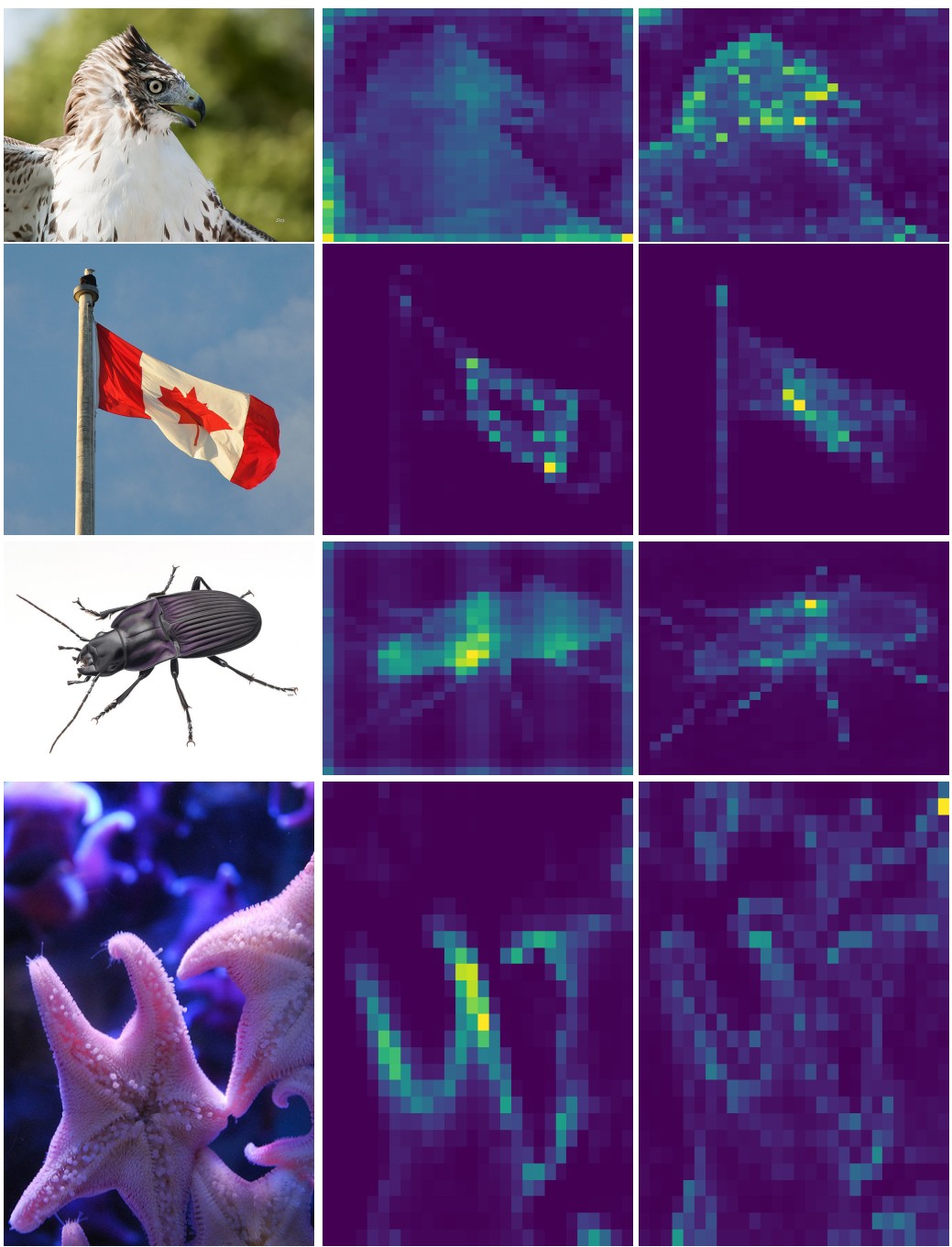

Figure 8: **Figure-ground segmentation in early layers of ViT/1**. In each row, we show the original image and the attention maps of two selected early layers (from first to fourth). We use the central pixel in the image space as the query and visualize its attention maps. Structures that can capture the foreground of objects have already emerged in these layers, which prepares for the later layers to learn higher-order relationships.

| epochs | model | FID (↓) | sFID (↓) | IS (↑) | precision (↑) | recall (↑) |
|--------|-------|---------|----------|--------|---------------|------------|
| 400 | DiT-L/2 | 4.16 | 4.97 | 210.18 | 0.88 | 0.49 |
| | **DiT-L/1** | **4.05** | **4.66** | **232.95** | **0.88** | **0.49** |
| 1400 | DiT-L/2 | 2.89 | 4.43 | 242.13 | 0.85 | 0.54 |
| | **DiT-L/1** | **2.68** | **4.34** | **268.82** | **0.85** | **0.55** |

Table 6: **Extended results for image generation.** We continue the training from 400 epochs (main paper) to 1400 epochs, and find the gap between DiT/2 and DiT/1 becomes larger, especially on FID.

## C   EXTENDED RESULTS ON IMAGE GENERATION

In the main paper (Sec. 4.3), both image generation models, DiT-L/2 and DiT-L/1, are trained for 400 epochs. To see the trend for longer training, we followed Peebles & Xie (2023) and simply continued training them till 1400 epochs while keeping the learning rate constant.

The results are summarized in Tab. 6. Interestingly, longer-training also benefits DiT/1 more than DiT. Note that FID shall be compared in a *relative* sense – a 0.2 gap around 2 is bigger than 0.2 around 4.

## D   TEXTURE *vs.* SHAPE BIAS ANALYSIS

As a final interesting observation, we used an external benchmark[3] which checks if an ImageNet classifier's decision is based on texture or shape. ConvNets are heavily biased toward texture (∼20 in shape bias). Interestingly, we find ViT/1 relies *more* on shape than ViT (57.2 *vs.* 56.7), suggesting that even when images are treated as sets of pixels, Transformers can still sift through potentially abundant texture patterns to identify and rely on the sparse shape signals for tasks like object recognition.

## E   ADDITIONAL NOTES FOR TRAINING TRANSFORMER ON PIXELS

While for some cases (*e.g.*, DiT (Peebles & Xie, 2023)), the training recipe can be directly transferred to pixel-based Transformer; for some other cases, we do want to note more potential challenges during training. Below we want to especially highlight the effect of *reduced learning rates* when training a supervised ViT from scratch.

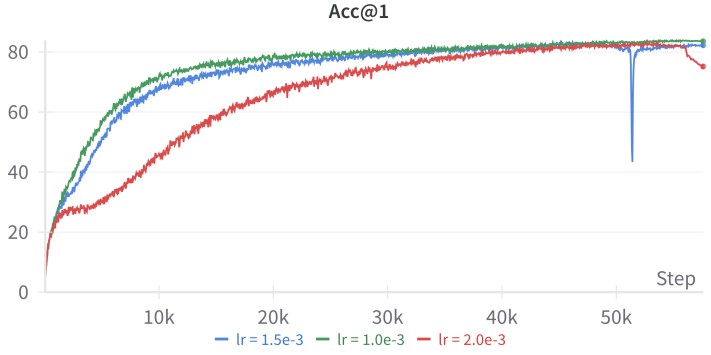

Figure 9: **Training curves of different learning rates** (**ViT-T/1**, batch size 1024, CIFAR-100). With a reduced learning rate from ViT/2, the curve is more stable, and ultimately leads to better accuracy.

We take CIFAR-100 as an example. As shown in Fig. 9, we find for pixel-based Transformer, the training becomes unstable if we maintain the same learning rate as vanilla ViTs. It is especially vulnerable toward the end. When the initial learning rate is reduced from $2e^{-3}$ to $1e^{-3}$, the training is more stable and leads to better accuracy. Similar observations are also made on ImageNet.

---

[3] https://github.com/rgeirhos/texture-vs-shape

| | dictionary | Acc@1 | Acc@5 |
|---|---|---|---|
| ViT-B/2 | | 75.7 | 92.3 |
| **ViT-B/1** | | 76.1 | 92.6 |
| | ✓ | **76.6** | **92.8** |

Table 7: Compared to the default ViT that uses parametric projection layers, we find a simple **dictionary-based ViT** that leverages the reduced vocabulary size to learn non-parametric embeddings for pixel intensity values works better for ImageNet supervised classification.

| | downsizing | Acc@1 | Acc@5 |
|---|---|---|---|
| ViT-B/2 | bicubic | 75.7 | 92.3 |
| **ViT-B/1** | | **76.1** | **92.6** |
| ViT-B/2 | single-pixel | 70.5 | 89.0 |
| **ViT-B/1** | | **75.6** | **92.3** |

Table 8: **Influence of downsizing algorithms** to our finding. Here we specifically experimented to downsize the original image by simply copying the value of its corresponding pixel from the original image ('single-pixel'), as opposed to 'bicubic interpolation' which leverages nearby pixels/locality. While single-pixel downsizing are generally worse, locality-free models are much more robust to such a change for their representations being locality-agnostic.

## F    EXPLORING DICTIONARY-BASED VIT

As discussed in Sec. 3, one advantage for treating individual pixels as tokens for ViT is the reduction in vocabulary size. Specifically, the size of the vocabulary (*i.e.*, the unique values the model must handle) becomes much more manageable.

To exploit this fact, we experimented with a *dictionary-based* approach for pixel tokens in ViT (other settings exactly follow default ViT for supervised learning). Specifically, we employed a *non-parametric*, 256-dimensional embedding for each of the 256 possible values a pixel can take, and different color channels have different embeddings. For a pixel with three (RGB) values, we mapped each value to its corresponding vector, and concatenated the three vectors. An optional linear projection was applied to ensure compatibility with the hidden dimension $d$ of ViT. Note that this approach has even less inductive bias because we no longer assume the relative order for pixel intensity values: the embedded value of 127 is not necessarily in-between 255 and 0, whereas it must be in-between if modeled directly with a parametric projection.

Tab. 7 summarizes the results. Interestingly, we find this ViT variant works better than our default ViT-B/1 on ImageNet. The improvement in performance further endorses our finding: 1) reducing inductive bias can be beneficial; and 2) locality-free ViTs can unlock the potential of dictionary-based token representations for next-generation vision architectures.

## G    INFLUENCE OF DOWNSIZING ALGORITHMS

By default, we use bicubic interpolation to perform downsizing. But since such algorithms also leverage nearby pixel values to estimate the value of the current pixel, there can be concerns that our pixel-based Transformers still rely on subtle locality biases.

To investigate the influence of downsizing algorithms, we have conducted additional experiments comparing bicubic interpolation with a simpler approach where each pixel in the downsized image *directly copies the value of its corresponding pixel from the original image*. In this way, it circumvents the locality bias introduced by interpolation. We call this downsizing algorithm 'single-pixel'.

The results for ImageNet classification, ViT-B are summarized in Tab. 8. Two observations are made. First, as expected, single-pixel downsizing results in lower performance compared to bicubic interpolation. But second and more interestingly, patch-based ViT-B/2 shows a more significant performance drop than the locality-free ViT-B/1, and the gap becomes larger. This suggests that locality-based models are more sensitive to this change in downsizing algorithm. In contrast, locality-free models are more robust, likely due to their ability to learn locality-agnostic representations and do not rely on the inductive bias of locality for classification.

| type of tokens | Acc@1 | Acc@5 |
|---|---|---|
| patchification ($16 \times 16$) | 82.7 | 96.3 |
| FH | 81.3 | 95.6 |
| SLIC (iter=2) | 78.9 | 94.3 |
| SLIC (iter=1) | 82.4 | 96.1 |

Table 9: An exploration of **advanced tokenizations**. Here, we find that the performance improves when moving away from irregular shape and closer to patchification.

## H EXPLORATION OF TOKENS VIA ADVANCED GROUPING

Before our journey with pixels as tokens, we were interested in tokens via advanced grouping (*i.e.*, SLIC (Achanta et al., 2010) and FH segmentation (Felzenszwalb & Huttenlocher, 2004)) due to its ability to quickly produce high-semantic and irregular shape tokens.

To compare these advanced tokenizations, we setup an experiment with the scratch training recipe from Touvron et al. (2021); He et al. (2022) with patch size being $16 \times 16$ and sequence length of 196. For FH segmentation, we use the implementation from scikit. The optimal FH scale (which indirectly controls the resulting sizes of tokens) is 30. We also set the maximum number of tokens to 225, as the number of tokens can vary across images. Different from FH which relies on graph-based grouping, SLIC is similar to k-means, so an important hyper-parameter is the number of iterations used to optimize the segments.

The results for ImageNet classification are shown in Tab. 9. The patchification ($16 \times 16$) performs best, even when compared against advanced tokens with additional training (e.g., FH does not converges as fast, so we used 400 epochs, longer than the default 300 epochs). SLIC (iter=1) performs close to patchification but visual inspection shows that it essentially reverts to patch-based tokens as part of the algorithm initialization. Further SLIC iterations, even with one more (iter=2), degrade performance. This suggests that the simple patchification is superior to more advanced grouping tokenization.

