# OpenReview forum: "An Image is Worth More Than 16x16 Patches: Exploring Transformers on Individual Pixels"
_ICLR.cc/2025/Conference — ICLR 2025 Poster_

### Official Review · Reviewer_Atm6 · 2024-11-03

**Soundness:** 2
**Presentation:** 3
**Contribution:** 2
**Rating:** 5
**Confidence:** 5

**Summary:**

The paper studies the role of locality biases in Vision Transformers (ViTs) by examining whether treating individual pixels as tokens (1x1 patches)—rather than conventional 16×16 patches—is sufficient for computer vision tasks (image classification or generation).

This pixel-based approach yields 2 benefits:
1. Reduced Vocabulary Size: By treating each pixel as an independent token, the "vocabulary" size (i.e., unique values the model must handle) becomes more manageable.
2. Model Agnosticism to Locality Bias: The pixel-based Transformer performs competitively across various tasks, such as image classification, self-supervised learning, and image generation, suggesting that locality may not be as essential for vision tasks for ViT as previously thought.

The authors evaluate this approach on tasks like classification (CIFAR-100, ImageNet), self-supervised learning, and image generation through diffusion models. They find that the pixel-based Transformer achieves results comparable to patch-based ViTs, though with higher computational costs due to increased sequence lengths.

While the paper is well-written, it falls short in addressing the practical implications of using 1x1 patches beyond merely demonstrating feasibility. (i.e. other than knowing that we can use 1x1 patches, what is the value of using 1x1 in practice?). I am hesitant to recommend this paper for NeurIPS.

**Strengths:**

* Clear and Detailed Writing: The paper is well-organized, with clear explanations of the rationale behind pixel-based tokenization, the potential benefits, and the authors are upfront about the trade-offs involved. I only suggest a few writing modifications below.

* Thorough Experimental Evaluation: The paper includes a comprehensive set of experiments across multiple tasks—classification, self-supervised learning, and image generation, although adding more baselines will make the evaluation more complete and convincing.

**Weaknesses:**

* The empirical results in Table 3 for Case Study #1 show only marginal performance improvements when using 1×1 tokens on datasets like ImageNet and Oxford-102-Flower. The evaluation would be more convincing if results were presented for all four size variants (ViT-T/S/B/L) and included a 16×16 baseline for comparison.
* In Case Study #2, there is a lack of detailed information on the setup and evaluation. It’s unclear what new value or insights this setup offers beyond those in Case Study #1. This section appears to have been completed in a rush.
* In Table 5, there is no clear advantage in using 1×1 tokens over 2×2 tokens. Results might benefit from additional baselines, such as 4×4 or 8×8 tokens.
* In Figure 3, qualitative examples for the 1×1 approach are shown without comparable examples from baseline methods. Including baseline results would provide a more effective comparison.
* While the authors are upfront about the inefficiency of using 1x1 patches, I am unsure about the practical value of using 1x1 patches (though vocab size benefit was mentioned). The overall takeaway for me is: “We can use 1x1 rather than 2x2 and we still get good performance” but the answer why this finding matters is not clearly explained.

**Questions:**

Rather than questions, I have a few suggestions to improve writing.

L411 Intrestingly, -> Interestingly

L201: in range [0,255],  the vocab size is 256^3 not 255^3. Please revise all the writing to ensure the correct information.

L44-46: The authors should clarify **why** translation equivariance still exists in ViT. Translation equivariance in neural networks, particularly in convolutional neural networks (CNNs), is the property where shifting the input (e.g., an image) results in a corresponding shift in the output features. This means the network recognizes that the content has moved spatially without altering the actual information content, which is useful for tasks like object detection and localization.

In transformers, translation equivariance is not naturally present. Transformers process input sequences as tokens, where each token attends to all other tokens through self-attention. transformers don’t inherently “understand” spatial shifts in the same way that CNNs do because each token’s positional information is encoded separately, often through positional embeddings.

All my concerns are written in Weaknesses section.

---

> ### Author Response · Authors · 2024-11-26
>
> Thank you for the comment recognizing our writing as clear and detailed, our paper as well-organized, and our experiments as thorough. Below, we address your concerns and have incorporated the relevant updates into our revised paper.
>
> > The empirical results in Table 3 for Case Study #1 shows only marginal performance improvements when using 1×1 tokens on datasets like ImageNet and Oxford-102-Flower. The evaluation would be more convincing if results were presented for all four size variants (ViT-T/S/B/L) and included a 16×16 baseline for comparison.
>
> Thank you for the suggestion. We have conducted additional experiments with ViT-L, and the results are summarized below:
>
> | model | acc@1 | acc@5 |
> |--------|--------|--------|
> | ViT-B/2 | 75.7 | 92.3 |
> | ViT-B/1 | **76.1** (+0.4) | **92.6** (+0.3) |
> | ViT-L/2 | 75.6 | 92.3 |
> | ViT-L/1 | **76.9** (+1.3) | **93.0** (+0.7) |
>
> Interestingly, ViT-L/2 shows signs of overfitting, with little improvement compared to ViT-B/2. However, ViT-L/1 demonstrates a larger performance gain, suggesting that locality-free designs scale more effectively with larger model sizes. These additional results are included in our revised draft, and we thank you for help improve our work!
>
> > In Case Study #2, there is a lack of detailed information on the setup and evaluation. It’s unclear what new value or insights this setup offers beyond those in Case Study #1. This section appears to have been completed in a rush.
>
> We apologize for any confusion caused by the omission of details in the main text. The full setup and evaluation details are provided in Section A of the appendix due to space limitations. To improve clarity, we have revised the main paper to explicitly reference the appendix for readers seeking additional information. We hope this resolves your concern.
>
> > In Table 5, there is no clear advantage in using 1×1 tokens over 2×2 tokens. Results might benefit from additional baselines, such as 4×4 or 8×8 tokens.
>
> Thank you for pointing this out. We have extended the results for image generation, as shown in Table 6 of our paper. According to the DiT paper, increasing token sizes to 4×4 or 8×8 results in *weaker* baselines, with FID scores degrading significantly (e.g., from 23.33 for 2x2 tokens to 45.64 for 4x4 and 118.87 for 8x8). Instead, in Table 6 we compare against a *stronger* baseline: DiT-L/2 trained for 1400 epochs.
>
> The results show that the gap between the locality-based DiT/2 and locality-free DiT/1 becomes *larger* especially on the main metric FID, and DiT/1 outperforms DiT/2 on 4 out of the 5 metrics (and tying on the fifth). We hope these additional results clarify the value of locality-free architectures for image generation.
>
> > In Figure 3, qualitative examples for the 1×1 approach are shown without comparable examples from baseline methods. Including baseline results would provide a more effective comparison.
>
> We appreciate this great suggestion! We have updated Figure 3 in the revised draft to include qualitative examples from DiT-L/2 for direct comparison. We believe this enhances the clarity and impact of our results, so thank you!
>
> > While the authors are upfront about the inefficiency of using 1x1 patches, I am unsure about the practical value of using 1x1 patches (though vocab size benefit was mentioned). The overall takeaway for me is: “We can use 1x1 rather than 2x2 and we still get good performance” but the answer why this finding matters is not clearly explained.
>
> Thank you for raising this point. To illustrate the practical value of 1×1 patches, we experimented with a dictionary-based approach for pixel tokens in ViT, leveraging the smaller vocabulary size. Specifically, we employed a non-parametric embedding of size $256 \times d$ for color channels. For a pixel with $[r, g, b]$ values, we mapped each value into a vector of dimension $d=256$, and concatenated the three vectors. An optional linear projection ensured compatibility with ViT’s hidden dimension.
>
> Interestingly, we find this dictionary-based ViT-B/1 works *better* than our ViT-B/1 on ImageNet:
> | model | acc@1 | acc@5 |
> |--------|--------|--------|
> | ViT-B/1 | 76.1 | 92.6 |
> | ViT-B/1, dictionary | **76.6** (+0.5) | **92.8** (+0.2) |
>
> This demonstrates two key points: 1) reduced inductive bias is beneficial, and 2) locality-free ViTs enable novel token representations, such as dictionary-based tokens, which “is essential for bridging the gap between computer vision and NLP” (as noted by PhHu).
>
> We have added these results in the updated draft (Sec. F). Thanks for the valuable suggestion to strengthen our work!

---

> ### Author Response · Authors · 2024-11-26
>
> > L44-46: The authors should clarify why translation equivariance still exists in ViT.
>
> This is great feedback that triggers us to think more carefully about this term. We followed the statement from the original ViT paper, which states “In ViT, only MLP layers are local and *translationally equivariant*, while the self-attention layers are global.” (https://arxiv.org/pdf/2010.11929, page 4, paragraph “inductive bias”). This statement is at the *operation-level*, limiting the discussion within certain layers.
>
> However, here the reviewer’s feedback is on the whole *architecture-level*, where ViTs viewed as holistic functions are indeed *not* translation equivariant. This is mostly because of the self-attention layers, but also because of different position embeddings appended to each location.
>
> On a side note, while ConvNets as an architecture are renowned for being translation equivariant, in practice they are often *not*, see for example the study in [2], and also for various location-sensitive tasks (e.g. object detection and segmentation), the network will leverage *zero-paddings* to keep track of the current location, which makes the final model not translation equivariant.
>
> As an attempt to make it more clear, we have decided to use *location equivariance* to refer to the *adoption of weight-sharing mechanism that ensures that the same weights are applied regardless of spatial locations*. We have thoroughly updated the draft to reflect this, with a footnote explaining what we mean when we say this. Please let us know your thoughts after reviewing the updates and if there are better ideas.
>
> Once again, we sincerely appreciate your detailed feedback, which has helped us strengthen our paper.
>
> [1] Dosovitskiy, Alexey. "An image is worth 16x16 words: Transformers for image recognition at scale." ICLR. 2021.
>
> [2] Zhang, Richard. "Making convolutional networks shift-invariant again." ICML. 2019.

---

> > ### Author Response · Authors · 2024-11-28
> >
> > > In the current version, the Figure 5 is mixed of both graph and table. Given the y-axes are similar, authors may consider aggregate the left and right or convert them into the same format.
> >
> > Thank you Atm6 for this additional feedback. We appreciate your suggestion regarding the format of Figure 5. After careful consideration, we will keep the current figure + table format to ensure clarity in conveying two distinct messages. Here is our reasoning:
> > - **Table (left)**: This focuses on the effect of varying the number of pixel swaps ($T$) while keeping the maximum distance $\delta$ fixed at its default value ($\infty$). The results highlight how pixel permutations alone can significantly reduce accuracy, emphasizing the importance of positional consistency in vision models.
> > - **Graph (right)**: This examines the joint variation of both $T$ and $\delta$, where the curves illustrate how allowing farther-away pixel swaps exacerbates the damage caused by permutations, providing a complementary perspective.
> >
> > While the y-axes are similar (ImageNet accuracy), the table and graph have different x-axes and deliver two separate messages. The table isolates the effect of $T$, while the graph focuses more on $\delta$. Our initial Fig 5 actually buried these two messages into a single plot, but this was less clear, leading us to adopt the current format.
> >
> > We have updated the Fig 5 caption to reflect this feedback, and hope this explanation clarifies our intent and rationale for keeping them separate.

---

> > ### Author Response · Authors · 2024-11-28
> >
> > > The authors mentioned an experiment regarding vocabulary size. Could you provide more details about the experimental setup to better assess its validity? Regarding the dictionary-based approach, what advantages does it offer beyond the observed accuracy improvement of +0.5 on acc@1?
> >
> > > I read thru Appendix Sec. F and still dont get how the experiment was done. I wish the authors could describe as detailed as possible.
> >
> > Thanks for the interest and feedback. We have outlined more descriptions below and hope they will provide a more comprehensive understanding of the dictionary-based ViT:
> > - **High-level**:
> > In this variant, we introduced *non-parametric embeddings* to represent pixels in ViT/1. Standard ViTs typically use fully *parametric* embeddings, where a linear layer is employed to project patches (or pixels, in the case of ViT/1) into embeddings. In contrast, the dictionary-based variant uses pre-defined embeddings for each possible pixel value.
> > - **Motivation beyond accuracy**:
> > While the observed improvement of +0.5 on acc@1 is notable, the primary motivation for this experiment was to demonstrate one of the key advantages of locality-free architectures: their ability to work with *reduced vocabulary sizes*, as also pointed out by the reviewer. This also aligns well with the broader narrative of our study, emphasizing reduction in inductive biases.
> > - **Inductive bias reduction**:
> > Specifically, the dictionary-based variant *does not assume a relative order* for pixel intensity values. For example, the embedding for a pixel intensity of 127 is not inherently “in-between” the embeddings for 255 and 0, as would be the case with parametric projections.
> > - **Experimental setups**:
> > In a nutshell, this experiment exactly followed the ImageNet supervised learning setup for ViT-B/1, with the only difference being the embedding mechanism. For further details, please refer to:
> >   - *Section 4.1*: Describes the dataset and evaluation settings;
> >   - *Appendix A (“Case Study #1,” second paragraph)*: Other details about ImageNet supervised learning;
> >   - *Appendix F*: Provides specific descriptions of the dictionary-based embedding variant.
> > - **Code release**:
> > We also plan to release the code to ensure that all details, including any not explicitly covered in the paper, are accessible to the community.
> >
> > If there are specific aspects that remain unclear, please let us know, and we will be happy to elaborate further.

---

> ### Author Response · Authors · 2024-11-28
> **Respond to justifications**
>
> Dear reviewer Atm6,
>
> Thanks for the updated thoughts and new feedback, we deeply appreciate your input and respect your perspective. Below we respond to each of your remaining comments:
>
> > There lacks the motivation for each of the use cases. Adding them into each Section (Use case) will make the writing/paper more solid.
>
> To clarify, we only have *one* motivation when presenting these use cases: to demonstrate the locality-free architecture -- Transformer on individual pixels works well. All the case studies are focused on this singular point. We have revised the leading paragraph of Sec 4 to make this overarching motivation explicit. This can be shared for all the case studies.
>
> > Other than demonstrating that 1x1 patches can be used, what is the value of using 1x1 patches?
>
> The most significant value lies in applications where *the concept of patches* does not exist. While vision applications often benefit from locality as a strong inductive bias, this is not universally true across domains. For instance, in tasks like modeling DNA sequences or protein structures, patches are not a natural construct. Demonstrating that locality can be removed in vision tasks suggests that existing models could potentially work well in such domains *without inducing specific biases*. While this border exploration is outside the scope of our study, we see this as a key implication of our findings beyond vision tasks.
>
> > Looking at the improvements over baselines in Tables, giving the info about statistical significance will clarify the advantage of using 1x1.
>
> The observed standard deviation in our experiments is consistently below 0.1% whenever we check. While it is common practice in our field not to report error bars for such small deviations (e.g., in baselines like ViT and DiT), we acknowledge that providing this context can be helpful for readers less familiar with the norms of the field. If needed, we can make a note of this in our revised draft.
>
> Additionally, we’d like to emphasize that our goal is to show that locality-free architectures can work well— and this does not necessarily mean to outperform locality-based methods by a significant margin. In fact, our extended efforts to explore advanced tokens repeatedly failed to even match patch-based approaches. The relative ease with which 1x1 tokens perform well conveys a valuable and surprising message worth sharing.
>
> > Also, in each Table, we see different versions of ViT being used as the competitors, giving impression of the results being cherry-picked. Could the authors help address this concern?
>
> The different ViT sizes reflect the datasets and computational constraints we faced during experimentation, rather than cherry-picking. For instance, we used smaller models for smaller datasets like CIFAR and larger ones (up to ViT-B and ViT-L) for larger datasets like ImageNet. The inclusion of ViT-L was specifically added in response to reviewers’ requests and shows good generalizability, but it was not tried due to our initial computational limitations.
>
> We are committed to transparency and take this “cherry-picking” concern seriously. While our original plan was not to release the code due to the simplicity of the modification, in light of this speculation, we now plan to release it to ensure full reproducibility and transparency.
>
> > The qualitative comparisons in Fig. 3 are not well done. Using the same prompts, we would expect the outputs of the two models to be directly comparable, highlighting clear contrasts and similarities. Currently, Fig.3 gives no more information.
>
> Thank you for this suggestion. We have updated Fig. 3 in the paper, ensuring that outputs use fixed random seeds and fixed category prompts. Although we still cannot fully control the output for a detail-aligned apple-to-apple comparison, the revised comparison is cleaner and more rigorous. We appreciate your input on this improvement.
>
> > In light of this, I am maintaining my current rating as a Reject. However, I will closely follow the discussion here, as I have not yet finalized my rating.
>
> We appreciate your willingness to engage in further discussion. While there are points where we respectfully disagree, your feedback has been highly valuable in refining our paper. We have responded to all concerns and updated our draft accordingly. Thank you for helping us improve our work and holding us to a high standard.

---

> > ### Comment · Reviewer_Atm6 · 2024-11-30
> >
> > > The most significant value lies in applications where the concept of patches does not exist
> >
> > I believe this claim is quite strong, and I cannot fully agree with it. Essentially, it contradicts the community's direction of leveraging more advanced tokens, which I believe is not the original message of this paper.
> >
> > > For instance, in tasks like modeling DNA sequences or protein structures, patches are not a natural construct. Demonstrating that locality can be removed in vision tasks suggests that existing models could potentially work well in such domains without inducing specific biases.
> >
> > Using an example from protein structures, which is not a vision-based task, to support the claim that "locality can be removed in vision tasks" seems misplaced and lacks relevance in this context.

---

> ### Author Response · Authors · 2024-12-01
>
> > I believe this claim is quite strong, and I cannot fully agree with it. Essentially, it contradicts the community's direction of leveraging more advanced tokens, which I believe is not the original message of this paper.
>
> We understand your concern and acknowledge that our claim challenges the current trajectory in the community of leveraging more advanced tokens. Prior to this project, we also shared the belief in this direction. However, through the challenges we faced in exploring advanced tokens and the consistent empirical evidence pointing to “pixels as tokens” as a more promising alternative, our perspective has evolved.
>
> In the earlier response (at reviewer NZzh), we clarified how our findings relate to the discussion on advanced tokens. Rather than dismissing the value of advanced tokens outright, our results suggest that in certain contexts (in fact more than we expected), simpler, locality-free designs may outperform approaches with higher inductive biases. We believe this shift in understanding is a key contribution of our work and an important message for the community to know.
>
> > Using an example from protein structures, which is not a vision-based task, to support the claim that "locality can be removed in vision tasks" seems misplaced and lacks relevance in this context.
>
> Thank you for highlighting this, and we agree that it would be misplaced to use examples from protein structures to support the claim that “locality can be removed in vision tasks.” To clarify, our reference to protein structures was intended to address the question, *“What is the value of using 1×1 patches?”*
>
> The key takeaway is that operating on 1×1 patches *effectively eliminates the concept of “patches” altogether*. This opens up the possibility of applying such models in domains where patches—or their equivalents—do not exist, such as DNA sequences or protein structures. While this broader applicability is outside the scope of our paper, it highlights *the potential value of our findings*.
>
> We position this work as a finding paper rather than a method paper, aiming to share insights about the flexibility and potential of locality-free architectures, particularly in contexts where traditional assumptions may not apply.
>
> We hope this clarifies our position and the intent behind the examples provided.

---

> > ### Comment · Reviewer_Atm6 · 2024-12-03
> > **My final ratings**
> >
> > Dear Authors,
> >
> > Thank you for your considerable effort in adding new experiments. I truly appreciate your hard work!!!
> >
> > That said, the entirely new experiments on "tokens via patchification" >> "tokens via advanced grouping" were introduced very late in the rebuttal process. Given the limited context and information provided (from paper and discussion here), I find it unable to properly/thoroughly evaluate this claim. In particular, the assertion that "tokens via patchification" >> "tokens via advanced grouping" is not convincingly substantiated in its current form.
> >
> > However, I do observe a clear trend: while 1x1 patchification demonstrates higher accuracy compared to tokens via patchification, it is significantly less efficient due to the computational overhead.
> >
> > I believe the paper would be more compelling if the authors could clearly demonstrate the trend: pixels as tokens >> tokens via patchification >> tokens via advanced grouping. Unfortunately, this claim is not sufficiently supported in its current state.
> >
> > Overall, this is a good finding paper, though I feel it offers limited practical values. I have raised my score from 3 to 5 (still leaning towards rejection) to acknowledge the authors' efforts, and I wish the ACs will carefully weigh my concerns in their final decision.

---

> > > ### Author Response · Authors · 2024-12-03
> > >
> > > Thank you for updating your rating, appreciating our work as “a good finding paper”, and carefully checking our results. While we do not expect further engagement, we feel responsible to respond while there is still time.
> > >
> > > > In particular, the assertion that "tokens via patchification" >> "tokens via advanced grouping" is not convincingly substantiated in its current form.
> > >
> > > Since these are *negative results*, we fully acknowledge it has limitations as one can argue that we haven't tried "hard enough" and may miss tricks to make advanced grouping work. This is precisely why these results were initially excluded, despite our early hopes that advanced grouping would succeed.
> > >
> > > > I do observe a clear trend: while 1x1 patchification demonstrates higher accuracy compared to tokens via patchification, it is significantly less efficient due to the computational overhead.
> > >
> > > We have been upfront about this computational limitation, framing our work as a *finding* paper rather than a *practical method* for immediate deployment. To us, the inefficiency is not an empirical trend but an inherent property of “pixels as tokens,” as higher FLOPs are required by design.
> > >
> > > That said, we do want to note:
> > > - **Advanced grouping with FH**: Despite setting the maximal number of tokens to 225 (>196 for $16\times16$ patches) and extending training length, FH still underperformed the baseline. This suggests that even more compute *does not necessarily* improve its performance.
> > > - **Potential value despite inefficiency**: Inefficiency can be justified if it helps achieve a higher accuracy upper bound. For instance, large language models (LLMs) today are far less efficient than previous-generation language models, yet their ability to unlock new capabilities makes them widely adopted -- and the research focus is *not* about how to stay with old models, but instead making LLMs more efficient.  Similarly, we believe our inefficient “pixels as tokens” as a finding can open new directions and opportunities for refinement.
> > >
> > > We hope the ACs will consider these points during the final decision-making process. Thank you again for your time and thoughtful feedback.

---

### Official Review · Reviewer_PhHu · 2024-11-04

**Soundness:** 4
**Presentation:** 4
**Contribution:** 2
**Rating:** 6
**Confidence:** 5

**Summary:**

This work poses an intriguing question regarding locality-free modeling of images using Transformers. The new modeling perspective centered on the locality-free concept presents an interesting direction for applying Transformers in computer vision, given their success in machine translation tasks. The study conducts extensive experiments across various scale tasks, revealing consistent and general observations. These findings lead to broader conclusions regarding the effectiveness of locality-free modeling under certain conditions.

**Strengths:**

1. The questions raised are interesting. The locality-free concept is essential for bridging the gap between computer vision and natural language processing when language models are adopted to solve visual recognition tasks.
2. The observations are compelling and support the general conclusions on the potentials of locality-free modeling under certain conditions. The study performs extensive experiments across various scales, revealing consistent and widespread findings.
3. The experiments are well designed to demonstrate the potential of locality-free modeling for images.

**Weaknesses:**

The conditions underlying the observations and conclusions require greater emphasis and detail. While "locality-free" suggests that input patches or tokens can be fragmented, it is still essential to preserve the relationships among adjacent input vectors or scalars. Developing a dictionary or tokenizer for image pixels is challenging, which hinders the advancement of locality-free models. The conclusions presented summarize valuable insights, but many of these observations, especially pixel-based modeling, probably have been considered during the design of Vision Transformers (ViTs). The limitations in their efficient training contribute to their performance issues. It is also quite challenging to represent image pixels semantically in a linguistic manner. If the authors could propose an effective training scheme or a pixel dictionary design for locality-free approaches, this work would become even more insightful and beneficial.

**Questions:**

The concept of locality-free in this work is conditional and needs to be clearly differentiated from random pixel-level training. All discussions regarding locality-free should specify these conditions; otherwise, the statements may be too bold. While the paper identifies a problem, it does not explore potential solutions or the feasibility of addressing them. The authors should address these questions during the rebuttal to improve the rating score.

---

> ### Author Response · Authors · 2024-11-26
>
> Thank you for the careful review and for acknowledging our observations as interesting and compelling, as well as recognizing our experiments as extensive and well-designed. We address concerns below and will incorporate them into our updated paper.
>
> > If the authors could propose an effective training scheme or a pixel dictionary design for locality-free approaches, this work would become even more insightful and beneficial
>
> Thanks for this great suggestion. Inspired by this comment, we experimented with a dictionary-based approach for pixel tokens in ViT. Specifically, we employed a non-parametric embedding of size $256 \times d$ for color channels. For a pixel with $[r, g, b]$ values, we mapped each value into a vector of dimension $d=256$, and concatenated the three vectors. An optional linear projection was applied to ensure compatibility with the hidden dimension of ViT.
>
> Interestingly, we find this dictionary-based ViT works *better* than our ViT-B/1 on ImageNet:
> | model | acc@1 | acc@5 |
> |--------|--------|--------|
> | ViT-B/1 | 76.1 | 92.6 |
> | ViT-B/1, dictionary | **76.6** (+0.5) | **92.8** (+0.2) |
>
> Note that this approach has *even less inductive bias* because we no longer assume the relative order for pixel intensity values (the embedded value of 127 is not necessarily in-between 255 and 0, whereas it must be in-between if modeled directly with a parametric projection).
>
> The improvement in performance further endorses our finding: 1) reduced inductive bias can be beneficial, and 2) locality-free ViTs can unlock the potential of dictionary-based token representations for vision. We have added these results in the updated draft (Sec. F).
>
> > many of these observations, especially pixel-based modeling, probably have been considered during the design of Vision Transformers (ViTs).
>
> While we cannot speak to the exact design trajectory behind ViT, one key difference between ViT (published at ICLR 2021) and its predecessor, iGPT (published at ICML 2020), lies in the tokenization strategy. iGPT employed pixel-level tokens but struggled with performance despite its high computational cost. In contrast, ViT reintroduced locality through patch-level tokens, achieving state-of-the-art results on benchmarks like ImageNet. This shift cemented 16×16 patches as the standard for vision tasks.
>
> However, prior to our study, it remained unclear whether ViT’s performance gains were primarily due to higher resolution or patch-based locality. Through systematic analysis, our work clarifies this distinction: resolution is the key enabler, while locality is not essential for strong performance. This finding has value, because it addresses an important mystery in the literature and sheds light on fundamental design principles for Transformers used in vision.
>
> > While the paper identifies a problem, it does not explore potential solutions or the feasibility of addressing them.
>
> Thank you for this feedback. We fully acknowledge that our work is focused on presenting a finding rather than proposing a new method. This decision was in fact intentional—we chose not to introduce or name a new locality-free architecture, instead giving full credit to existing models like ViT. By doing so, we aimed to clearly highlight our message to the community without overshadowing it with additional methodological contributions. We believe this trade-off was necessary to convey the significance of our findings and their implications for future research.
>
> Once again, we sincerely appreciate your thoughtful comments, which have helped us refine and improve our work.

---

> > ### Comment · Reviewer_PhHu · 2024-11-27
> > **Reply to authors' comments**
> >
> > I appreciate the authors' clarification, as most of the responses are very helpful in understanding this work. However, my only concern is that this is a findings paper, offering interesting observations without providing a specific solution.
> > Therefore, I would be slightly inclined to recommend this paper for acceptance; however, if it is rejected due to the lack of a solution, it would not be detrimental.

---

> > > ### Author Response · Authors · 2024-11-27
> > >
> > > Thanks PhHu for the updated thoughts. We standby our work as a findings paper, and give full credits to existing methods like ViT/DiT.
> > >
> > > Should you have more questions and comments, we are happy to respond to them before the deadline.

---

### Official Review · Reviewer_5fCb · 2024-11-04

**Soundness:** 3
**Presentation:** 3
**Contribution:** 2
**Rating:** 8
**Confidence:** 4

**Summary:**

This paper presents a study of locality in ViT, through decreasing the patch size down to 1x1. The paper shows that smaller patch will always get better performance and 1x1, which does not encode locality in each patch, gets the best performance. The work contributes to understandings of whether locality / patchification is necessary. The authors conduct experiments on supervised learning, self-supervised learning, and image generation. The results validate the hypotheses and statements.

**Strengths:**

+ The paper is well-written and has great clarity on motivation and experiment designs.
+ The authors extensively test the model on various tasks and benchmarks.
+ It's interesting to see that the patchification design to be explicitly re-checked.

**Weaknesses:**

- The work focuses on using smaller side of ViTs. I wonder if scaling up, does 1x1 patch still help in performance? Could it be that for ViT/L or ViT/H the patchification does not matter that much among 1x1, 2x2, or 4x4?

- This paper proposes a finding, but it seems no good solution is available for now. The growth of #tokens is quadratic, 512 x 512 will give you 262144, and 1048576 for 1024 x1024. Further more, video will even give more tokens. Flash Attention and other efficient version will still have a tough time taking these many tokens as inputs.

- The authors discuss a series of related works. Another important and emerging line of work that uses grouping operation is missing though. Images as set of points[1] is strongly relevant work that breaks the image into 1x1 pixel representation and uses fixed centers clustering / pooling for representation learning. The slot attention-based backbone [2], although do not use 1x1 patches, made the grouping operator (instead of key-axis attention in ViT) work on ImageNet scale and should also be discussed. Do the authors think grouping can be a more handy way for forming "flexible and irregular-shaped patches" automatically, and enjoy the sweet spot of lower computation while having smaller patch sizes?

[1] Image as Set of Points

[2] Perceptual Group Tokenizer: Building Perception with Iterative Grouping

**Questions:**

See weakness.

---

> ### Author Response · Authors · 2024-11-26
>
> Thank you for the careful review and detailed comments. We greatly appreciate your recognition of our work as well-written, with great clarity in our motivation and experiment design, as well as the acknowledgment of our extensive experiments. We address the remaining concerns below.
>
> > The work focuses on using the smaller side of ViTs. I wonder if scaling up, does 1x1 patch still help in performance? Could it be that for ViT/L or ViT/H the patchification does not matter that much among 1x1, 2x2, or 4x4?
>
> Thanks for the valuable feedback. To explore the impact of scaling up, we trained ViT-L on ImageNet using both pixel tokens (/1) and patch tokens (/2), following the scratch training recipe from the MAE paper. The results are summarized below:
>
> | model | acc@1 | acc@5 |
> |--------|--------|--------|
> | ViT-B/2 | 75.7 | 92.3 |
> | ViT-B/1 | **76.1** (+0.4) | **92.6** (+0.3) |
> | ViT-L/2 | 75.6 | 92.3 |
> | ViT-L/1 | **76.9** (+1.3) | **93.0** (+0.7) |
>
> Interestingly, ViT-L/2 shows signs of overfitting, with stagnation in both accuracies compared to ViT-B/2. In contrast, ViT-L/1 demonstrates a healthy improvement, suggesting that locality-free designs not only scale effectively but also yield greater benefits as model sizes increase. These findings have been included in our updated draft, and we thank you for prompting this investigation – it greatly enhances our work!
>
> > This paper proposes a finding, but it seems no good solution is available for now. The growth of # of tokens is quadratic, 512 x 512 will give you 262144, and 1048576 for 1024 x1024. Furthermore, video will even give more tokens. Flash Attention and other efficient versions will still have a tough time taking these many tokens as inputs.
>
> We fully agree with this limitation, which we have explicitly discussed in the paper. On the other hand, this challenge underscores the significance of our finding. Beyond scaling models and datasets—two common strategies in vision and language modeling—our work highlights sequence length scaling as a viable scaling direction, *even at the expense of locality*. Yes there are technical challenges, but our work is to raise the awareness of this potential direction, and believe this is valuable for driving innovation in this area.
>
> > Another important and emerging line of work that uses grouping operation is missing though. Images as a set of points … slot attention-based backbone … Do the authors think grouping can be a more handy way for forming "flexible and irregular-shaped patches" automatically, and enjoy the sweet spot of lower computation while having smaller patch sizes?
>
> This is an insightful suggestion. Interestingly, we began this project by exploring flexible and irregular-shaped grouping methods, hoping to improve performance while reducing sequence length. We devoted considerable time to this approach, experimenting with techniques ranging from basic clustering algorithms like k-means to more advanced methods such as FH [1], SLIC [2], and SAM [3]. Unfortunately, none of these methods demonstrated performance comparable to patch-based ViTs.
>
> Based on our explorations and the fact that no one has demonstrated more advanced tokens can work as well as patches, our current answer is that grouping methods, while conceptually appealing, have yet to show practical promise in this context. If desired, we can add a brief discussion of these approaches as related work. However, we believe their limited effectiveness does not significantly impact our findings. Please let us know if you think otherwise.
>
> Thank you again for your valuable feedback, which has further enriched our study.
>
> [1] Felzenszwalb, Pedro F., and Daniel P. Huttenlocher. "Efficient graph-based image segmentation." IJCV. 2004.
>
> [2] Achanta, Radhakrishna, et al. "SLIC superpixels." 2010.
>
> [3] Kirillov, Alexander, et al. "Segment anything." ICCV. 2023.

---

> ### Comment · Reviewer_5fCb · 2024-11-26
> **Thank you for the response**
>
> I'd like to thank the authors for the response. The experiment part addresses my concern.
>
> Two more questions:
>
> (1) I wonder whether the authors have explored the masking ratios when patch size is set to 1x1? Does masking ratio go by number of tokens (i.e. larger number of tokens needs higher masking ratio) or it depends on the specific dataset?
>
> (2) On ImageNet, does the performance advantage of 1x1 patch still hold after finetuning the MAE trained models? Current results are focused on linear probings.
>
> For the related work, yes please add all those five papers (or more) to make it more complete. Grouping methods might still need other engineering recipes but offering that direction as an option will be beneficial to both readers and the field.

---

> > ### Author Response · Authors · 2024-11-27
> >
> > Thanks for the prompt response and engagement in improving our work!
> >
> > >  I wonder whether the authors have explored the masking ratios when patch size is set to 1x1? Does masking ratio go by number of tokens (i.e. larger number of tokens needs higher masking ratio) or it depends on the specific dataset?
> >
> > We haven't. We adopted the same mask ratio for 1x1 patch size and it already works well in the initial shot, so we did not explore further.
> >
> > Intuitively, it makes sense that larger number of tokens (or rather, more delicate tokens) would call for higher mask ratio, as even a small number of visible tokens can *leak* information for reconstruction -- which in turn hurts representation learning. We note one practical way to effectively maintain the mask ratio when changing the sequence length is to *decouple* the patch size used for masking from the patch size used for Transformers [1], but we find in our experiments for 1x1 patch size on CIFAR, it's not necessary.
> >
> > > For the related work, yes please add all those five papers (or more) to make it more complete. Grouping methods might still need other engineering recipes but offering that direction as an option will be beneficial to both readers and the field.
> >
> > Thanks for the feedback and requests. We have added a small paragraph discussing about the mentioned work and others (and made a pass to other parts of the related work to fit in the space). Indeed it would be helpful for interested readers.
> >
> > Again, we appreciate all the feedback and suggestions, and believe our revised submission has significantly improved in quality. If further modifications are needed, we would be happy to make them happen before the deadline. Thanks!
> >
> > [1] Hu, Ronghang, et al. "Exploring long-sequence masked autoencoders." arXiv preprint arXiv:2210.07224 (2022).

---

> > > ### Author Response · Authors · 2024-11-27
> > >
> > > For the added second question:
> > >
> > > > On ImageNet, does the performance advantage of 1x1 patch still hold after finetuning the MAE trained models? Current results are focused on linear probings.
> > >
> > > Sorry for the confusion, but all the results we have reported with MAE are with fine-tuning on CIFAR; there is no linear probing involved. We choose fine-tuning as the evaluation protocol as this is the biggest strength of MAE pre-training. Hope it clarifies the confusion.

---

> ### Comment · Reviewer_5fCb · 2024-11-27
> **Thank you. Finalizing my review.**
>
> Thank you for all the above responses.
>
> I do think this paper provides useful information, especially given the current status that vision backbones have stagnated for a long time. Significance might be measured differently across reviewers. For me, I think the paper is significant enough to be accepted. **I was hoping to improve the rating to 7** while unfortunately there's no such rating in this year's ICLR.
>
> I do have more requests to make this paper better - cifar experiments are great, but even ImageNet2012 nowadays is a bit cute for experiments and analysis. Some findings might vary and may not hold when scaling happens. It would be really nice if the authors can add MAE SSL ImageNet results, with and without finetuning, or even DINO + 1x1 patch experiments and analysis. These will make the conclusions stronger and more convincing.
>
> I'll respect other reviewer or ac's opinion as well if theirs differs.

---

> ### Author Response · Authors · 2024-11-27
>
> Thanks 5fCb for the updated opinion, your input has greatly enhanced our work.
>
> We fully agree more experiments are important to strengthen the signal. That's why after our initial finding on CIFAR, we scaled it to ImageNet for supervised learning and generative modeling as verifications. For MAE, it adds variety to the experiment in terms of *tasks*, but we could not afford both task- and dataset-generalization back then.
>
> We most likely won't have time to get the ImageNet MAE results before the draft-updating deadline, but we will strive to make an update about this before the discussion period ends.

---

> ### Author Response · Authors · 2024-12-03
>
> > We most likely won't have time to get the ImageNet MAE results before the draft-updating deadline, but we will strive to make an update about this before the discussion period ends.
>
> An update on MAE pre-trained ViT-B on ImageNet with 28$\times$28 input:
>
> | model | pre-train | acc@1 | acc@5 |
> |--------|--------|--------|--------|
> | ViT-B/2 | no | 75.7 | 92.3 |
> | ViT-B/1 | no | **76.1** (+0.4) | **92.6** (+0.3) |
> | ViT-B/2 | yes | 76.0 | 92.5 |
> | ViT-B/1 | yes | **76.5** (+0.5) | **92.9** (+0.4) |
>
> The results are consistent with our finding that 1x1 patch size (or Transformer on pixels) works well. MAE pre-training also helps.
>
> We will add these results to the updated draft.

---

### Official Review · Reviewer_NZzh · 2024-11-09

**Soundness:** 3
**Presentation:** 4
**Contribution:** 2
**Rating:** 6
**Confidence:** 5

**Summary:**

The study investigates whether the inductive bias of locality is necessary for modern computer vision models. It explores the effectiveness of using vanilla Transformers to directly treat each individual pixel as a token for various computer vision tasks. The findings show that pixel-based Transformers can perform competitively in supervised and self-supervised learning, as well as image generation, challenging the belief that locality is essential. The research examines how removing locality impacts architectures like Vision Transformers (ViT) and compares results with patch-based models. Although computationally intensive, the study demonstrates that locality-free Transformers can still learn effective visual representations, which may inform future neural network designs.

**Strengths:**

1. This paper is well-written.
2. The topic is of great interest to the computer vision community.
3. The experiments are extensive.

**Weaknesses:**

In general, I agree to recommend the acceptance of this paper. However, I think it is a borderline paper, due to the following reasons.
1. The finding is not surprising: removing the locality of visual backbones, and the models still work, at least to some extent. I think this conclusion can be expected by most of the researchers within the community (btw, I appreciate the efforts of authors to make these observations clear).
2. As also recognized by the authors, the practicality of this conclusion is questionable, given its limited computation efficiency.
3. The down-sampling algorithm (e.g., bicubic resampling) has already introduced some inductive biases of locality. It can be seen as a convolutional layer with fixed kernels. This effect may be particularly notable when the resolution is relatively small (e.g., 28x28 on ImageNet).

**Questions:**

See Weaknesses.

---

> ### Author Response · Authors · 2024-11-26
>
> Thank you for the detailed and constructive feedback on our work. We appreciate the recognition of our topic’s relevance to the computer vision community and your acknowledgment of the quality of our paper and the breadth of our experiments. Below, we address your concerns.
>
> > The finding is not surprising: removing the locality of visual backbones, and the models still work, at least to some extent. I think this conclusion can be expected by most of the researchers within the community (btw, I appreciate the efforts of authors to make these observations clear).
>
> We sincerely thank you for acknowledging our efforts to clarify our findings. To address the “surprisingness” concern, we would like to share the research trajectory behind this work. Initially, our project aimed to use SAM [1] as an advanced tokenizer for visual backbones, an idea supported by concurrent efforts exploring similar directions. This approach seemed intuitive, as many researchers believed SAM tokens could outperform standard 16x16 patches. However, after months of exploration, we found that advanced tokens consistently degraded performance instead of improving it. This motivated our pivot toward a simpler yet unconventional approach: breaking patches into even smaller units—pixels. Surprisingly, this pivot led to better results within just a few weeks.
>
> Given the prevalence of the belief that more advanced tokens would outperform rudimentary ones, we respectfully disagree with the notion that our conclusion was expected. In fact, our findings challenge the widely held assumption that locality is fundamental for vision models, representing a conceptual shift from this belief. While our modification can seem *practically* small, *conceptually* we have gone through a long journey to overcome the mental bias and stop believing locality is essential for vision models.
>
> We hope this clarifies our perspectives and appreciate your understanding.
>
> > As also recognized by the authors, the practicality of this conclusion is questionable, given its limited computation efficiency
>
> Thanks for the comment. We fully acknowledge this limitation, as increasing the token length inherently demands more computation, especially when paired with self-attention mechanisms. However, we remain optimistic that advancements in engineering (e.g., Ring Attention [2]) and scientific innovations (e.g., Mamba [3]) will make locality-free models more computationally feasible in the future.
>
> > The down-sampling algorithm (e.g., bicubic resampling) has already introduced some inductive biases of locality. It can be seen as a convolutional layer with fixed kernels. This effect may be particularly notable when the resolution is relatively small (e.g., 28x28 on ImageNet).
>
> Thank you for highlighting this important point. To investigate the influence of downsizing algorithms, we have conducted additional experiments comparing bicubic interpolation with a simpler approach where each pixel in the downsized image *directly copies the value of its corresponding pixel from the original image*, thereby removing the locality bias introduced by interpolation.
>
> The results for ImageNet, ViT-B are summarized below:
> | method | model | acc@1 | acc@5 |
> |--------|--------|--------|--------|
> | bicubic |  ViT-B/2 | 75.7 | 92.3 |
> | bicubic |  ViT-B/1 | **76.1** (+0.4) | **92.6** (+0.3) |
> | single pixel |  ViT-B/2 | 70.5 | 89.0 |
> | single pixel |  ViT-B/1 | **75.6** (+5.1) | **92.3** (+3.3) |
>
> As expected, single-pixel downsizing results in lower performance compared to bicubic interpolation. But interestingly, patch-based ViT-B/2 shows a more significant performance drop than the locality-free ViT-B/1 (so the gap becomes larger), suggesting that locality-dependent models are more sensitive to this change. In contrast, locality-free models demonstrate robustness here, likely due to their ability to learn locality-agnostic representations and do not rely on the inductive bias of locality.
>
> We have incorporated the results in our draft (Sec. G).
>
> We hope these additional analyses and explanations address your concerns. Thank you again for your valuable feedback, which has helped us further refine our work.
>
> [1] Kirillov, Alexander, et al. "Segment anything." ICCV. 2023.
>
> [2] Liu, Hao, Matei Zaharia, and Pieter Abbeel. "Ring attention with blockwise transformers for near-infinite context." arXiv:2310.01889.
>
> [3] Gu, Albert, and Tri Dao. "Mamba: Linear-time sequence modeling with selective state spaces." arXiv:2312.00752.

---

> ### Author Response · Authors · 2024-11-27
>
> > Can you provide the references for prevalence of the belief that more advanced tokens would outperform rudimentary ones. This will show if your findings are expected or not as suggested by NZzh.
>
> Thank you for the comment, Atm6. In our updated draft, we have included references to four recent works that explore advanced tokens designed to intuitively outperform patch tokens (partially thanks to the pointers from 5fCb). These references are as follows:
>
> [1] Xiao Zhang and Michael Maire. Self-supervised visual representation learning from hierarchical grouping. In NeurIPS, 2020. *Spotlight*
>
> [2] Tsung-Wei Ke, Jyh-Jing Hwang, Yunhui Guo, Xudong Wang, and Stella X Yu. Unsupervised hierarchical semantic segmentation with multiview cosegmentation and clustering transformers. In CVPR, 2022. *Oral*
>
> [3] Xu Ma, Yuqian Zhou, Huan Wang, Can Qin, Bin Sun, Chang Liu, and Yun Fu. Image as set of points. In ICLR, 2023. *Oral*
>
> [4] Zhiwei Deng, Ting Chen, and Yang Li. Perceptual group tokenizer: Building perception with iterative
> grouping. In ICLR, 2024.
>
> Additionally, follow-up works, such as the one below we just found, further demonstrate the sustained interest in this direction:
>
> [5] Tsung-Wei Ke, Sangwoo Mo, and Stella X Yu. Learning hierarchical image segmentation for recognition and by recognition. In ICLR, 2024. *Spotlight*
>
> While we are confident there are more references we could include with additional time for a broader survey (need to focus on the final draft and experiments now), we hope these examples are sufficient to demonstrate the belief within the community that advanced tokens would outperform rudimentary ones. Note that not only there is a line of work published in top-tier venues, many of which received *Oral* or *Spotlight* presentations, indicating strong reviewer appreciation and high ratings.
>
> To sum up our current position on this matter:
>
> - **Community Interest**. As referenced above, many researchers have interest and belief in the potential of advanced tokens, as demonstrated by their work.
> - **Informal Discussions**. While we do not have hard records, in our communications with researchers, this topic frequently arises (e.g., as questions or answers), especially following the release of SAM, which has sparked further interest in token design.
> - **Acknowledgment of Language**. Sure, “prevalence” may be too strong a word to describe the community’s belief. We are happy to rephrase this. However, based on the above evidence, we believe it is reasonable to conclude that many researchers had good faith in the potential of advanced tokens -- and will continue pushing the frontier for this direction.
>
> Given this context, we believe our finding is indeed surprising. It challenges the assumption that advanced tokens are inherently better and suggests the *opposite*: the most rudimentary tokens (e.g., pixels) can serve as effective inputs, allowing powerful architectures like Transformers to ***learn** to group implicitly* for vision tasks.
>
> We hope this clarifies the concern and are open to revisiting or expanding on this during the discussion period if needed.

---

> ### Comment · Reviewer_Atm6 · 2024-11-30
> **respectfully disagree with your claims**
>
> > While we are confident there are more references we could include with additional time for a broader survey (need to focus on the final draft and experiments now), we hope these examples are sufficient to demonstrate the belief within the community that advanced tokens would outperform rudimentary ones.
>
> > Given this context, we believe our finding is indeed surprising. It challenges the assumption that advanced tokens are inherently better and suggests the opposite: the most rudimentary tokens (e.g., pixels) can serve as effective inputs, allowing powerful architectures like Transformers to learn to group implicitly for vision tasks.
>
> Thank you for your detailed response. However, I must respectfully `disagree` with your reasoning.
>
> Here’s why:
>
> - In your rebuttal and paper, you demonstrate that 1x1 tokens outperform 2x2 patches.
>
> - Then, you also provide examples of advanced tokens from recent works (references [1-5]), which represent sophisticated tokenization strategies designed to improve performance.
>
> Given this, I do not see how 2x2 naive patches can be equated to the advanced tokens described in your references [1-5]. The comparisons made in your study seem to focus on rudimentary patches (e.g., 1x1 vs. 2x2), **yet you claim the findings challenge the belief in advanced tokens**.
>
> Without a direct comparison between your approach and truly advanced tokens (as highlighted in references [1-5]), the conclusion that rudimentary tokens (e.g., 1x1 patches) outperform advanced tokens appears unsupported. Simply comparing naive 2x2 patches to 1x1 tokens and framing this as a novel and surprising finding overlooks the broader context of advanced tokenization methods.

---

> ### Author Response · Authors · 2024-12-01
>
> > Given this, I do not see how 2x2 naive patches can be equated to the advanced tokens described in your references [1-5]. The comparisons made in your study seem to focus on rudimentary patches (e.g., 1x1 vs. 2x2), yet you claim the findings challenge the belief in advanced tokens.
>
> Thanks for the feedback. We appreciate the opportunity to clarify further.
>
> The underlying assumption in the community, as evidenced by references [1-5] and other related works, can be summarized as follows:
>
> **tokens via advanced grouping $\gg$ tokens via patchification $\gg$ pixels as tokens,**
>
> where $\gg$ denotes expected better performance. Note that advanced grouping incorporates both *geometry* and *content*; patchification relies solely on *geometry* without incorporating content; and pixels as tokens lack both geometry and content information, representing the least amount of inductive bias used.
>
> However, our empirical studies demonstrate a *completely reversed* performance order:
>
> **pixels as tokens $\gg$ tokens via patchification $\gg$ tokens via advanced grouping.**
>
> This reversal is surprising (and at very least, interesting), particularly because we ourselves initially shared the belief that advanced grouping would outperform simpler tokenization strategies. We believe the contrast between the observed performance and the assumed one *underscores* the value of our finding.
>
> Our paper focused on the *positive aspect* of this finding: pixels as tokens $\gg$ tokens via patchification. However, we understand your concern and agree that the *negative results*—tokens via patchification $\gg$ tokens via advanced grouping—are also worth noting for providing a complete picture. We can include related results in the appendix of the revised draft for further transparency and context (cannot edit the draft now).
>
> We hope this addresses your concern and clarifies the reasoning above.

---

> > ### Comment · Reviewer_Atm6 · 2024-12-01
> >
> > Dear authors,
> >
> > Thank you for the time and great efforts!
> >
> > I have a concern about for the following claim.
> >
> > > However, our empirical studies demonstrate a completely reversed performance order
> >
> > Could you please point me to where did you show the reversed trend? currently the main message is 1x1 > 2x2 (and 1x1 is not significantly better as denoted by your `>>` or `<<`).
> >
> > If the authors could show the data for:
> >
> > - pixels as tokens
> >
> > - tokens via patchification
> >
> > - tokens via advanced grouping.
> >
> > and the reversed trend, I will be convinced that the findings are not expected (or interesting).

---

> ### Author Response · Authors · 2024-12-02
>
> Thanks for the active follow-up Atm6. We really appreciate the time and opportunity.
>
> We can provide pointers/data in *two* parts, the *positive aspect of the finding* presented in the paper and *negative results* from our exploratory experiments. We start with the *positive finding*: "pixels as tokens" $\gg$ "tokens via patchification".
>
> - In the paper, 1$\times$1 tokens represent "pixels as tokens", while $p\times p$ ($p>1$) represent
>  "tokens via patchification".
> - The results across *Table 3 (a,b,c,d), Figure 2, Table 4 (a, b), Table 5, Table 6, Table 7, Table 8* compare these two types of tokens.
> - We primarily focused on $p=2$ as this is the *strongest* patchification baseline, but Figure 2 also illustrates the trend when the patch size is increased further.
> - In response to your request for larger patch sizes (e.g., $4\times4$, $8\times8$) in Table 5, we just obtained results for DiT-L models with 400-epoch training::
> | model | FID$\downarrow$ | IS$\uparrow$ |
> |--------|--------|--------|
> | DiT-L/2 | 4.16 | 210.18 |
> | DiT-L/1 | **4.05** | **232.95** |
> | DiT-L/4 | 9.23 | 116.11 |
> | DiT-L/8 | 47.32 | 34.51 |
> - The first two rows are from the paper, and the last two are new results. Notably, our results for large patches are much better than those reported in the original DiT paper (e.g., FID$\downarrow$: 9.23 vs. 45.64 for DiT-L/4). Despite this improvement, the trend remains consistent with our findings: “pixels as tokens” outperform “tokens via patchification.”
> - We are happy to include these additional results in the appendix for transparency.
>
> We also went back and checked the *negative results* from our explorations —"tokens via patchification" $\gg$ "tokens via advanced grouping". The experiments are conducted on ImageNet, for supervised learning with ViT-B and $224\times 224$ inputs. Several methods are compared:
> - **Patchification baseline**. We used the scratch training recipe from [1,2], with patch size being $16\times 16$, and sequence length $196$.
> - **FH [3]**. This is the first grouping-based tokens we tried. We used the implementation from scikit (https://scikit-image.org/). The optimal FH scale (which indirectly controls the resulting sizes of tokens) is 30, and we set the maximal number of tokens to 225, as the number of tokens can vary across images.
> - **SLIC [4]**. This is the second advanced token. Different from FH which relies on graph-based grouping, SLIC is similar to k-means, so an important hyper-parameter is the number of iterations used to optimize the segments.
> - We report the following results:
> | type of tokens | acc@1 | acc@5 |
> |--------|--------|--------|
> | $16\times 16$ patches | **82.7** | **96.3** |
> | FH | 81.3 | 95.6 |
> | SLIC, iter$=2$ | 78.9 | 94.3 |
> | SLIC, iter$=1$ | 82.4 | 96.1 |
> Several observations are made:
> - $16\times16$ patches perform best, even when compared against advanced tokens with additional training (e.g., FH does not converges as fast, so we used 400 epochs, longer than the default 300 epochs).
> - SLIC with iter$=1$ performs close to patchification but visual inspection shows that it *essentially reverts to patch-based tokens* as part of the algorithm initialization. Further SLIC iterations, even with *one more* (iter$=2$), degrade performance.
> - We also experimented with SAM [5] tokens. The training was prohibitively slow, as the deep learning based SAM model was applied online for each image after cropping. And early results were *even worse* than SLIC so we did not wait till the training to fully finish. While we lack final numbers for SAM, the preliminary evidence also supports the trend that as an even more advanced token candidate, SAM is no better than SLIC.
>
> We can put these results and associated discussions in the appendix of the updated draft, if reviewers find it valuable ($\gg$ may be an exaggeration for the performance gap we have observed, $>$ or $\ge$ may be more appropriate).
>
> We hope these additional results and explanations address your concerns. If you feel further evidence or clarification is needed, please let us know before the discussion period ends, and we will do our best to provide it.
>
> [1] Touvron, Hugo, et al. "Going deeper with image transformers." ICCV. 2021.
>
> [2] He, Kaiming, et al. "Masked autoencoders are scalable vision learners." CVPR. 2022.
>
> [3] Felzenszwalb, Pedro F., and Daniel P. Huttenlocher. "Efficient graph-based image segmentation." IJCV. 2004.
>
> [4] Achanta, Radhakrishna, et al. "SLIC superpixels." 2010.
>
> [5] Kirillov, Alexander, et al. "Segment anything." ICCV. 2023.

---

> > ### Comment · Reviewer_Atm6 · 2024-12-03
> > **My final ratings**
> >
> > Dear Authors,
> >
> > Thank you for your considerable effort in adding new experiments. I truly appreciate your hard work!!!
> >
> > That said, the entirely new experiments on "tokens via patchification" >> "tokens via advanced grouping" were introduced very late in the rebuttal process. Given the limited context and information provided (from paper and discussion here), I find it unable to properly/thoroughly evaluate this claim. In particular, the assertion that "tokens via patchification" >> "tokens via advanced grouping" is not convincingly substantiated in its current form.
> >
> > However, I do observe a clear trend: **while 1x1 patchification demonstrates higher accuracy compared to tokens via patchification, it is significantly less efficient due to the computational overhead**.
> >
> > I believe the paper would be more compelling if the authors could clearly demonstrate the trend: pixels as tokens >> tokens via patchification >> tokens via advanced grouping. Unfortunately, this claim is not sufficiently supported in its current state.
> >
> > Overall, this is a good finding paper, though I feel it offers limited practical values. I have raised my score from 3 to 5 (still leaning towards rejection) to acknowledge the authors' efforts, and I wish the ACs will carefully weigh my concerns in their final decision.

---

> ### Author Response · Authors · 2024-12-02
> **Reminder for reviewer NZzh**
>
> Dear reviewer NZzh,
>
> It's been several days since we posted our response. Please let us know if you had a chance to read it and have an updated thought. If anything is less clear, we are happy to address it before the discussion period ends.

---

> > ### Comment · Reviewer_NZzh · 2024-12-03
> >
> > I really appreciate the efforts of the authors during the discussion. I have carefully read the responses and the comments from other reviewers. In fact, I do not think the authors have fully addressed my concerns on 'the finding is not surprising'.
> >
> > It may be quite questionable to purely talk about 'performance' in the absence of considering efficiency, especially in the community of computer vision (I think this may be common sense). 1x1 patch may lead to higher accuracy, due to its fine-grained representation and its significantly increased computation. However, 16x16 patch, for example, may also utilize the same amount of computation, for higher performance, as well. I think two questions may be important.
> > 1. With the same amount of computation, will 1x1 or larger patch be better?
> > 2. If a sufficient amount of computation and a sufficiently large model are available, will 1x1 or larger patch be better?
> >
> > I still think that the findings of this paper may not be surprising enough. Hence, I decide to keep my score.

---

> ### Author Response · Authors · 2024-12-03
>
> Thank you for updating your rating, appreciating our work as “a good finding paper”, and carefully checking our results. While we do not expect further engagement, we feel responsible to respond while there is still time.
>
> > In particular, the assertion that "tokens via patchification" >> "tokens via advanced grouping" is not convincingly substantiated in its current form.
>
> Since these are *negative results*, we fully acknowledge it has limitations as one can argue that we haven't tried "hard enough" and may miss tricks to make advanced grouping work. This is precisely why these results were initially excluded, despite our early hopes that advanced grouping would succeed.
>
> > I do observe a clear trend: while 1x1 patchification demonstrates higher accuracy compared to tokens via patchification, it is significantly less efficient due to the computational overhead.
>
> We have been upfront about this computational limitation, framing our work as a *finding* paper rather than a *practical method* for immediate deployment. To us, the inefficiency is not an empirical trend but an inherent property of “pixels as tokens,” as higher FLOPs are required by design.
>
> That said, we do want to note:
> - **Advanced grouping with FH**: Despite setting the maximal number of tokens to 225 (>196 for $16\times16$ patches) and extending training length, FH still underperformed the baseline. This suggests that even more compute *does not necessarily* improve its performance.
> - **Potential value despite inefficiency**: Inefficiency can be justified if it helps achieve a higher accuracy upper bound. For instance, large language models (LLMs) today are far less efficient than previous-generation language models, yet their ability to unlock new capabilities makes them widely adopted -- and the research focus is *not* about how to stay with old models, but instead making LLMs more efficient.  Similarly, we believe our inefficient “pixels as tokens” as a finding can open new directions and opportunities for refinement.
>
> We hope the ACs will consider these points during the final decision-making process. Thank you again for your time and thoughtful feedback.

---

> ### Author Response · Authors · 2024-12-03
>
> Thanks NZzh for the comment and the updated thoughts. While we do not expect further engagement, we feel responsible to address the efficiency concern you raised.
>
> > It may be quite questionable to purely talk about 'performance' in the absence of considering efficiency, especially in the community of computer vision (I think this may be common sense). 1x1 patch may lead to higher accuracy, due to its fine-grained representation and its significantly increased computation. However, 16x16 patch, for example, may also utilize the same amount of computation, for higher performance, as well.
>
> We agree that efficiency is an important consideration, and we have been upfront about this limitation in the paper. On the other hand, we have added two more notes: 1) on "advanced grouping with FH" where more compute doesn't automatically translate to better performance; and 2) on "potential value despite inefficiency" to share our forward-looking perspective. Please see the above responses to Atm6.
>
> > With the same amount of computation, will 1x1 or larger patch be better?
>
> > If a sufficient amount of computation and a sufficiently large model are available, will 1x1 or larger patch be better?
>
> We believe Figure 2 provides clear answers:
> - Figure 2a: With roughly the same amount of computation, 1x1 patches are consistently *the worst*, addressing the first question.
> - Figure 2b: With sufficient computation, the trend *reverses*—1x1 patches consistently become *the best*, answering the second question.
>
> These contrasting trends in Figure 2 highlight why this finding has been overlooked previously. Limited computational budgets favor patch-based methods, but when compute is abundant, 1x1 patches emerge as the superior choice for maximizing performance (regardless of efficiency). Our conclusion, therefore, is that with infinite compute, 1x1 should always be the go-to option for achieving the performance "upper bound".
>
> We hope this illustrates the value of our paper as one for "good findings" (that provide clear answers to important questions like these), and clarifies the concern.

---

### Author Response · Authors · 2024-11-26
**Shared response from authors**

We sincerely thank all the reviewers for their time, effort, and thoughtful feedback. We appreciate the recognition of several key aspects of our work, as highlighted across the reviews:

- **Significance**: “The topic is of great interest to the computer vision community” (NZzh); “The locality-free concept is essential for bridging the gap between computer vision and natural language processing” (PhHu)
- **Novelty**: "It's interesting to see that the patchification design to be explicitly re-checked” (5fCb); “The questions raised are interesting” (PhHu)
- **Writing**: “This paper is well-written” (NZzh); “The paper is well-written and has great clarity on motivation and experiment designs” (5fCb); “Clear and Detailed Writing” (Atm6)
- **Experiments**: “The experiments are extensive” (NZzh); “The authors extensively test the model on various tasks and benchmarks” (5fCb); “The observations are compelling and support the general conclusions” (PhHu); The study performs extensive experiments across various scales, revealing consistent and widespread findings” (PhHu); “The experiments are well designed” (PhHu); Thorough Experimental Evaluation (Atm6)

We have carefully addressed each reviewer’s comments and questions individually, and we have made significant revisions to the draft to incorporate the feedback. We hope our responses adequately address all outstanding concerns.

For convenience, below we summarize our revisions to the draft, all high-lighted in blue:

1. **Added** experiments on larger models (ViT-L) in Sec. 4.1 and Tab. 3;
1. **Added** experiments with dictionary-based ViT in Sec. F and Tab. 7;
1. **Added** analysis of the downsizing algorithm in Sec. G and Tab. 8;
1. **Added** discussions to related work on flexible, learnable patches via grouping in Sec. 6, per request by 5fCb;
1. **Revised** Fig. 3 to show better qualitative comparisons between DiT/2 and DiT/1;
1. **Defined** what we mean by “location equivariance” in footnote 1, aimed to clarify the confusion of “translation equivariance”;
1. **Added** clarifications for motivation in Sec. 4;
1. **Added** explicit reference to implementation details for all the case studies in Sec. 4;
1. **Fixed** the vocabulary size counting mistake, as pointed out by Atm6;
1. **Clarified** the messages from Fig. 5 for both the table and the figure;
1. **Addressed** typo in L411, as pointed out by Atm6.

We are grateful for the reviewers’ constructive input, which has significantly strengthened our work. If further clarifications are needed, we would be happy to provide them before the deadline.

---

### Meta-Review · Area_Chair_Rgd5 · 2024-12-21

**Metareview:**

This paper explores whether inductive bias towards locality, a hallmark of convolutional networks and Vision Transformers (ViTs), is truly necessary for computer vision tasks. The study replaces patch-based tokenization (e.g., 16x16 patches) with a "pixels-as-tokens" approach, where each individual pixel is treated as a token. The results demonstrate that this locality-free design performs competitively across various tasks, including supervised learning, self-supervised learning, and image generation. However, this approach is computationally intensive, owing to the increased sequence length.

Strengths:
* Challenging status of quo: The work challenges a fundamental belief in computer vision about the importance of locality and provides a new perspective for Transformer-based vision models.
* Thorough Experiments: Extensive evaluations across tasks and datasets add credibility to the findings.
* Clarity: The paper is well-written, with clear explanations of the motivation, experimental setup, and results.

Weaknesses:
* Practical Limitations: The approach is computationally inefficient due to the quadratic growth of tokens, making it less practical for large-scale vision tasks.
* Expected Findings: Some reviewers felt the results were not surprising, given prior knowledge that reducing inductive bias often leads to competitive results in deep learning.
* Late Introduced Claims: Some claims, such as comparisons between patchification and advanced tokenization, were introduced late in the rebuttal process and lacked sufficient substantiation.

Despite concerns about practicality and broader significance, this paper provides a meaningful contribution to the field by challenging established assumptions about locality in vision models. Its findings, though not entirely unexpected, are supported by robust experiments and could inspire further innovations in designing Transformers for vision tasks and beyond.

**Additional Comments On Reviewer Discussion:**

Surprisingness of Findings (NZzh, Atm6): Reviewers questioned the novelty and significance of the findings, suggesting that removing locality was not surprising.
* Response: Authors clarified the novelty of the reversal in performance trends and provided additional context to counter assumptions of expected results. They included comparisons between advanced tokens and patchification to strengthen the argument.

Practical Value (Atm6, NZzh): Concerns about the computational inefficiency of 1x1 tokens and lack of clear practical implications.
* Response: Authors acknowledged inefficiency but framed the work as a findings paper. They emphasized its conceptual importance for domains without locality and demonstrated potential applications with dictionary-based embeddings.

Experimental Robustness (Atm6): Concerns about statistical significance, cherry-picking baselines, and clarity in qualitative comparisons.
* Response: Authors added experiments for ViT-L, updated Figure 3 for clearer qualitative comparisons, and committed to code release for transparency. They also provided additional results and context to address statistical concerns.

Advanced Token Comparisons (Atm6, NZzh): Limited exploration of advanced tokens and late introduction of negative results in rebuttals.
* Response: Authors shared additional results on advanced tokenization methods (e.g., FH, SLIC) and acknowledged the limitations of these experiments.

Clarity and Presentation (PhHu, Atm6): Reviewers suggested improvements in motivation for use cases, figure formats, and writing precision.
* Response: Authors revised figures, clarified motivations, and corrected textual inconsistencies, including discussions on locality and translation equivariance.

---

### Decision · Program_Chairs · 2025-01-22

Accept (Poster)